# Kernel Density Steering: Inference-Time Scaling via Mode Seeking for Image Restoration

**Yuyang Hu**[1,2]*, **Kangfu Mei**[1], **Mojtaba Sahraee-Ardakan**[1],
**Ulugbek S. Kamilov**[2], **Peyman Milanfar**[1], **Mauricio Delbracio**[1],
[1]Google, [2]Washington University in St. Louis
{kangfumei,mojtabaa,milanfar,mdelbra}@google.com
{h.yuyang,kamilov}@wustl.edu,

## Abstract

Diffusion models show promise for image restoration, but existing methods often struggle with inconsistent fidelity and undesirable artifacts. To address this, we introduce Kernel Density Steering (KDS), a novel inference-time framework promoting robust, high-fidelity outputs through explicit local mode-seeking. KDS employs an $N$-particle ensemble of diffusion samples, computing patch-wise kernel density estimation gradients from their collective outputs. These gradients steer patches in each particle towards shared, higher-density regions identified within the ensemble. This collective local mode-seeking mechanism, acting as "collective wisdom", steers samples away from spurious modes prone to artifacts, arising from independent sampling or model imperfections, and towards more robust, high-fidelity structures. This allows us to obtain better quality samples at the expense of higher compute by simultaneously sampling multiple particles. As a plug-and-play framework, KDS requires no retraining or external verifiers, seamlessly integrating with various diffusion samplers. Extensive numerical validations demonstrate KDS substantially improves both quantitative and qualitative performance on challenging real-world super-resolution and image inpainting tasks.

## 1 Introduction

Image restoration (IR) seeks to recover a high-quality image $x$ from its degraded observation $y$. Recently, diffusion models (DMs) [1–3] have been successfully applied to challenging IR tasks [4–8], including super-resolution (SR) [9–14], image deblurring [15–17], and image inpainting [18–20]. DMs operate on a dual process: a forward process introduces noise to a clean image $x$ over T timesteps, creating a sequence of noisy latents $z_t$ where $z_T$ is approximately pure noise. A learned reverse process then iteratively refines these latents from $z_T$ back to an estimate of $z_0$.

To enable DMs for IR task, a straightforward way is to train diffusion models directly conditioned on information $c$ derived from the low-quality image $y$ [21–23]. Conditioning can be implemented through various techniques, for example, input concatenation [21], cross-attention [24, 25], adapters like ControlNet [26] or LoRA [27]. This allows the model's network $\epsilon_\theta(z_t, t, c)$ to estimate the conditional score function $\nabla_{z_t}\log p_t(x_t|c)$ to directly approximate target score function $\nabla_{z_t}\log p_t(z_t|y)$, enabling standard sampling at inference.

A key challenge here lies in navigating the perception-distortion trade-off [28]. Standard posterior sampling can yield perceptually sharp results, but often suffers from high distortion and hallucinations. Conversely, averaging multiple independent samples approximates the Minimum Mean Squared Error (MMSE) estimate, minimizing average distortion but leading to blurriness and poor perceptual

---

*This work was done during an internship at Google.

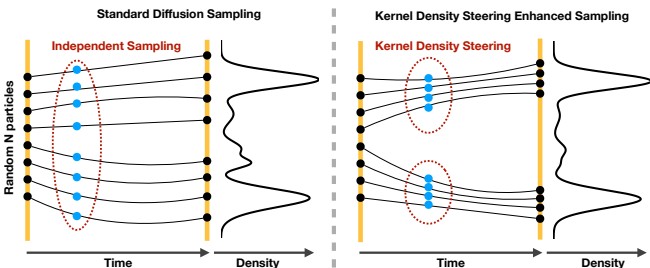

Figure 1: Conceptual illustration of Kernel Density Steering (KDS) for diffusion sampling. Left: Standard diffusion sampling with $N$ independent particles (blue dots) can result in high variance. Right: KDS utilizes the ensemble of $N$ particles to estimate local density. It then guides these particles towards shared high-density modes (peaks in the density curve) during the diffusion process.

quality [29]. Neither extreme is ideal for high-quality IR. Many existing methods improve this trade-off by additional model training or specialized network architectures [4, 29–34].

This paper introduces **Kernel Density Steering (KDS)**, an inference-time framework designed to improve the distortion-perception trade-off without requiring any model retraining. KDS leverages collective information from an $N$-particle ensemble by steering particles towards regions of high concentration within the predicted solution ensemble.

This mechanism is implemented as a two-step process. First, at each step $t$, the denoiser predicts the clean data $\hat{\mathbf{z}}_0^{(i)}$ for each particle $i$. Second, it performs a non-parametric mode-seeking operation to steer each prediction $\hat{\mathbf{z}}_0^{(i)}$ toward a local region of high density. This is achieved by implicitly estimating the ensemble's density using Kernel Density Estimation (KDE) [35, 36] and then applying a single mean-shift update—which is equivalent to a gradient ascent step on this estimated density—to find a consensus point $\tilde{\mathbf{z}}_0^{(i)}$:

$$
\tilde{\mathbf{z}}_0^{(i)} = \frac{\sum_{k=1}^{N} K\left(\frac{\|\hat{\mathbf{z}}_0^{(i)} - \hat{\mathbf{z}}_0^{(k)}\|^2}{h^2}\right) \hat{\mathbf{z}}_0^{(k)}}{\sum_{k=1}^{N} K\left(\frac{\|\hat{\mathbf{z}}_0^{(i)} - \hat{\mathbf{z}}_0^{(k)}\|^2}{h^2}\right)}. \tag{1}
$$

This refined estimate $\tilde{\mathbf{z}}_0^{(i)}$ defines a new consensus score, $\tilde{\mathbf{s}}_t^{(i)}$. The original and consensus scores are defined as:

$$
\mathbf{s}_t^{(i)} = \frac{\mathbf{z}_t^{(i)} - \sqrt{\bar{\alpha}_t}\hat{\mathbf{z}}_0^{(i)}}{\sigma_t^2} \quad \text{and} \quad \tilde{\mathbf{s}}_t^{(i)} = \frac{\mathbf{z}_t^{(i)} - \sqrt{\bar{\alpha}_t}\tilde{\mathbf{z}}_0^{(i)}}{\sigma_t^2}. \tag{2}
$$

This guidance is incorporated into the probability flow ODE via a weighted average of the two scores:

$$
d\mathbf{z}_t^{(i)} = \left[f(t)\mathbf{x}_t^{(i)} - \frac{1}{2}g(t)^2\left((1-\delta_t)\mathbf{s}_t^{(i)} + \delta_t\tilde{\mathbf{s}}_t^{(i)}\right)\right] dt, \tag{3}
$$

where $\delta_t$ is the KDS guidance strength. Thus, particles are guided by the original score and the KDS term. This term functions as a form of collaborative filtering, using non-parametric mode-seeking to guide the ensemble toward areas of high consensus. This adaptive, internal guidance aims for restorations that are both quantitatively accurate and perceptually sharp.

To demonstrate the effectiveness of this approach, we first evaluate KDS's impact on posterior sampling in a controlled synthetic task. Subsequently, we provide extensive empirical validation, showing significant improvements in both quantitative performance and qualitative robustness on challenging real-world SR and inpainting tasks.

**Our contributions are: (1)** We introduce Kernel Density Steering (KDS), a novel ensemble-based, inference time guidance method that uses patch-wise KDE gradients to steer diffusion sampling towards high-density posterior modes. **(2)** KDS improves the distortion-perception trade-off in image restoration by operating directly on the sampling process without retraining. **(3)** We design a patch-wise mechanism to overcome the curse of dimensionality, enabling KDS to scale to high-dimensional latent space. **(4)** Empirical validation showing significant improvements in fidelity, perception, and robustness on challenging real-world super-resolution and inpainting tasks.

## 2 Background

### 2.1 Latent Diffusion Models

Latent Diffusion Models (LDMs) [24] efficiently perform image generation by operating in a lower-dimensional latent space learned by a pre-trained autoencoder [37], mitigating the computational cost of pixel-space methods on high-resolution images. The autoencoder includes an encoder $\mathcal{E}$ mapping an image $\boldsymbol{x}$ to a latent $\boldsymbol{z} = \mathcal{E}(\boldsymbol{x})$, and a decoder $\mathcal{D}$ mapping $\boldsymbol{z}$ back to $\hat{x} = \mathcal{D}(\boldsymbol{z})$.

The forward diffusion process gradually adds Gaussian noise to the target latent sample $\boldsymbol{z}_0 = \mathcal{E}(\boldsymbol{x}_0)$ over $T$ timesteps according to a noise schedule $\alpha_t$. This process results in the following marginal distribution for the latent $\boldsymbol{z}_t$ at any timestep $t$:

$$q(\boldsymbol{z}_t|\boldsymbol{z}_0) := \mathcal{N}(\boldsymbol{z}_t; \sqrt{\bar{\alpha}_t}\boldsymbol{z}_0, (1 - \bar{\alpha}_t)\mathbf{I}), \text{where } \bar{\alpha}_t = \prod_{s=1}^{t} \alpha_s. \tag{4}$$

The reverse process aims to generate a clean latent variable $\boldsymbol{z}_0$ from a noisy latent variable $\boldsymbol{z}_T \sim \mathcal{N}(0, \mathbf{I})$. This is achieved by training a neural network $\boldsymbol{\epsilon}_\theta(\boldsymbol{z}_t, t)$ to predict the noise component $\boldsymbol{\epsilon}_t$ that was added during the forward process. This predicted noise is directly related to the score function of the noisy latent distribution $p_t(\boldsymbol{z}_t)$: $\nabla_{\boldsymbol{z}_t}\log p_t(\boldsymbol{z}_t) \approx -\frac{1}{\sqrt{1-\bar{\alpha}_t}}\boldsymbol{\epsilon}_\theta(\boldsymbol{z}_t, t)$. By accurately predicting the noise, the network learns to estimate this score function, thereby implicitly capturing the data distribution $p(\boldsymbol{z}_t)$ at different noise levels.

To sample from the learned distribution, Denoising Diffusion Implicit Models (DDIM) [38] offer a fast, non-Markovian sampling procedure. The sampling step from $\boldsymbol{z}_t$ to $\boldsymbol{z}_{t-1}$ in DDIM, which predicts the latent state at $t - 1$ based on the estimated clean latent at time $t$, is given by:

$$\boldsymbol{z}_{t-1} = \sqrt{\bar{\alpha}_{t-1}}\left(\frac{\boldsymbol{z}_t - \sqrt{1 - \bar{\alpha}_t}\boldsymbol{\epsilon}_\theta(\boldsymbol{z}_t, t, \boldsymbol{c})}{\sqrt{\bar{\alpha}_t}}\right) + \sqrt{1 - \bar{\alpha}_{t-1}}\boldsymbol{\epsilon}_\theta(\boldsymbol{z}_t, t, \boldsymbol{c}). \tag{5}$$

At the end of sampling, the predicted $\boldsymbol{z}_0$ is then decoded by $\mathcal{D}$ to obtain the generated image $\hat{\boldsymbol{x}}_0$.

### 2.2 LDM for Image restoration

Two primary methods exist for leveraging diffusion models in IR. Conditional LDMs [4–6, 9–11, 15, 16, 18–20, 39], learn the posterior $p(\boldsymbol{z}|\boldsymbol{c})$ directly, where the conditioning $\boldsymbol{c}$ is derived from $\boldsymbol{y}$. These models can be highly effective but may face challenges in accurately modeling the true posterior for complex degradations, potentially impacting restoration fidelity. Alternatively, unconditional LDMs can be employed with inference-time guidance using Bayes' rule [40–45]. These methods typically require computing gradients of the log-likelihood $\log p(\boldsymbol{y}|\boldsymbol{x}_t)$ (often approximated via the estimate $\hat{\boldsymbol{x}}_{0|t}$). This necessitates knowledge of the degradation process (e.g., the operator $A$ in $\boldsymbol{y} \approx A\boldsymbol{x}$), limiting their applicability in scenarios where the degradation is unknown or cannot be accurately modeled which is the case in many real-world image SR or deblurring.

### 2.3 Inference-Time Scaling in Diffusion Models

Merely increasing the number of denoising steps (NFEs) for a single sample yields diminishing returns [46], motivating research into more effective inference-time strategies [47, 48]. One prominent avenue, particularly explored in Text-to-Image (T2I) generation, involves actively searching for optimal initial noise vectors ($\boldsymbol{z}_T$) or resampling generation trajectories, often guided by external verifiers (e.g., CLIP score, pre-trained classifiers) to build a reward function and select promising candidates [47, 48]. However, this introduces reliance on these external verifiers, which can suffer from biases or misalignment with the true image restoration (IR) target.

A second class of strategies are specifically designed for imaging inverse problems. These often employ ensemble methods, such as those based on Sequential Monte Carlo (SMC) techniques [49–52], employ likelihood approximation and reweighting sampling trajectories to achieve more accurate approximation of the posterior distribution $p(\boldsymbol{x}|\boldsymbol{y})$ with unconditional DMs. These approaches, however, require accurate knowledge of the degradation model specific to the IR task. This assumption is often invalid in many real-world applications, such as real-world image SR and deblurring.

In contrast to both these approaches, KDS introduces an ensemble method that operates without external verifiers or the need for the knowledge of degradation model. By focusing on internal

consensus guidance derived directly from particle interactions within the ensemble, KDS offers a more robust and broadly applicable solution for enhancing image restoration quality.

## 2.4 Mean Shift for Mode Seeking

Mean Shift [53, 54] is a non-parametric clustering and mode-seeking algorithm. Given a set of points $\{\boldsymbol{x}_k\}_{k=1}^{N}$, the goal is to find the modes (local maxima) of the underlying density function. The algorithm iteratively shifts each point towards the local mean of points within its neighborhood, weighted by a kernel function (often Gaussian). For a point $\boldsymbol{x}$ and a kernel $K$ with bandwidth $h$, the mean shift vector is calculated as:

$$\mathbf{m}(\boldsymbol{x}) = \frac{\sum_{k=1}^{N} K\left(\frac{\|\boldsymbol{x}-\boldsymbol{x}_k\|^2}{h^2}\right) \boldsymbol{x}_k}{\sum_{k=1}^{N} K\left(\frac{\|\boldsymbol{x}-\boldsymbol{x}_k\|^2}{h^2}\right)} - \boldsymbol{x}. \tag{6}$$

A crucial property of the mean shift vector, $\mathbf{m}(\boldsymbol{x})$, is its direct proportionality to the gradient of the logarithm of the Kernel Density Estimate (KDE): $\mathbf{m}(\boldsymbol{x}) \propto \nabla_{\boldsymbol{x}} \log \hat{p}_K(\boldsymbol{x})$. Therefore, iteratively updating a point by adding its mean shift vector ($\boldsymbol{x} \leftarrow \boldsymbol{x} + \mathbf{m}(\boldsymbol{x})$) is an effective method for performing gradient ascent on the KDE, guiding the point towards a local mode of the estimated density. This established connection is leveraged in our proposed KDS framework.

# 3 Kernel Density Steering for Diffusion Samplers

Kernel Density Steering (KDS) is an inference-time technique for Image Restoration that enhances diffusion model sampling by leveraging an $N$-particle latent ensemble, $\boldsymbol{Z}_t = \{\boldsymbol{z}_t^{(i)}\}_{i=1}^{N}$. Standard diffusion sampling can result in outputs with inconsistent fidelity and undesirable artifacts. KDS addresses this by guiding all particles towards regions of high consensus within their own ensemble which is hypothesized to lead to more robust and high-fidelity restorations.

KDS achieves this ensemble-driven mode-seeking by incorporating an additional steering term into the sampling update for each particle. This steering is inspired by the Mean Shift algorithm (Sec. 2.4). Specifically, for each particle, KDS computes a mean shift vector $\mathbf{m}(\boldsymbol{z}_t^{(i)})$ based on the current ensemble $\boldsymbol{Z}_t$. Then KDS steering is applied by taking a step in the direction of this mean shift vector, which directs to a local mode of the ensemble's Kernel Density Estimate (KDE).

## 3.1 Patch-wise Mechanism and Integration

Directly applying KDE and mean shift in the full, high-dimensional latent space $\boldsymbol{z}_t$ is not achievable. A prohibitively large number of particles would be needed to obtain a reasonable density estimate via KDE in such high-dimensional spaces. To address this, we introduce a patch-wise mechanism to apply the KDS principle to smaller, more manageable, lower-dimensional patches. These patches are extracted from the predicted clean latent state $\hat{\boldsymbol{z}}_{0|t}^{(i)}$ corresponding to each particle $\boldsymbol{z}_t^{(i)}$ in the ensemble. This predicted clean latent, $\hat{\boldsymbol{z}}_{0|t}^{(i)}$, is the output of the diffusion model's denoising step at time $t$, obtained via:

$$\hat{\boldsymbol{z}}_{0|t}^{(i)} = \frac{1}{\sqrt{\bar{\alpha}_t}} \left( \boldsymbol{z}_t^{(i)} - \sqrt{1-\bar{\alpha}_t} \boldsymbol{\epsilon}_\theta(\boldsymbol{z}_t^{(i)}, t, \boldsymbol{c}) \right). \tag{7}$$

**Patch-wise Mechanism:** To the ensemble of predicted clean latent states $\{\hat{\boldsymbol{z}}_{0|t}^{(i)}\}_{i=1}^{N}$, our patch-wise mechanism involves the following steps at each timestep $t$:

1. *Patch Extraction:* For each predicted clean latent $\hat{\boldsymbol{z}}_{0|t}^{(i)}$ in the ensemble (obtained from Eq. 7), extract a set of non-overlapped patches $\{\boldsymbol{p}_j^{(i)}\}_j$ where we have omitted the time index to simplify the notation. This yields an ensemble of corresponding patches $\boldsymbol{P}_j = \{\boldsymbol{p}_j^{(k)}\}_{k=1}^{N}$ for each given spatial patch location $j$.

2. *Compute and Apply Patch-wise Steering:* For each patch $\boldsymbol{p}_j^{(i)}$ (from particle $i$ at location $j$), its mean shift vector $\mathbf{m}(\boldsymbol{p}_j^{(i)})$ is first computed based on the local ensemble of corresponding patches

$\boldsymbol{P}_j = \{\boldsymbol{p}_j^{(k)}\}_{k=1}^N$. This vector, which directs the patch towards a region of higher consensus within the ensemble, is given by:

$$\mathbf{m}(\boldsymbol{p}_j^{(i)}) = \frac{\sum_{k=1}^N G\left(\frac{\|\boldsymbol{p}_j^{(i)} - \boldsymbol{p}_j^{(k)}\|^2}{h^2}\right) \boldsymbol{p}_j^{(k)}}{\sum_{k=1}^N G\left(\frac{\|\boldsymbol{p}_j^{(i)} - \boldsymbol{p}_j^{(k)}\|^2}{h^2}\right)} - \boldsymbol{p}_j^{(i)}. \tag{8}$$

Here, $G$ is a Gaussian kernel function and $h$ is the bandwidth hyperparameter. As discussed in Sec. 2.4, this vector $\mathbf{m}(\boldsymbol{p}_j^{(i)})$ points from the current patch towards the estimated local mode of its ensemble's density. This computed mean shift vector is then used to update the patch:

$$\hat{\boldsymbol{p}}_j^{(i),\text{KDS}} \leftarrow \boldsymbol{p}_j^{(i)} + \delta_t \mathbf{m}(\boldsymbol{p}_j^{(i)}), \tag{9}$$

where $\delta_t \in [0,1]$ is a time-dependent steering strength.

3. *Reconstruct Guided Latent Prediction:* After all patches $\{\boldsymbol{p}_j^{(i)}\}_j$ from a given $\hat{\boldsymbol{z}}_{0|t}^{(i)}$ have been updated to $\{\hat{\boldsymbol{p}}_j^{(i),\text{KDS}}\}_j$, they are reassembled by direct replacement to original coordinate to form the KDS-refined clean latent prediction, $\hat{\boldsymbol{z}}_{0|t}^{(i),\text{KDS}}$.

**Integration with Diffusion Samplers:** KDS integrates seamlessly with standard diffusion samplers (e.g., first-order ODE DDIM solver and high-order ODE DPM-Solver++ [55]). At each sampling step $t$, the sampler first computes the predicted clean latent for each particle, $\hat{\boldsymbol{z}}_{0|t}^{(i)}$, using (Eq. 7). Patch-wise KDS is then applied to this ensemble of predictions $\{\hat{\boldsymbol{z}}_{0|t}^{(i)}\}_{i=1}^N$ to produce the ensemble of refined predictions $\{\hat{\boldsymbol{z}}_{0|t}^{(i),\text{KDS}}\}_{i=1}^N$. The sampler subsequently uses these KDS-refined estimates to compute the next latent states $\{\boldsymbol{z}_{t-1}^{(i)}\}_{i=1}^N$. After the full reverse diffusion process, an ensemble of KDS-guided clean latent states $\{\hat{\boldsymbol{z}}_0^{(i),\text{KDS}}\}_{i=1}^N$ is obtained.

**Final Particle Selection:** To produce a single, representative output image from an ensemble of generated candidates $\{\hat{\boldsymbol{z}}_0^{(i),\text{KDS}}\}_{i=1}^N$, a selection is made in the latent space. Our proposed strategy is to choose the latent vector from the ensemble that is closest to the ensemble's mean $\bar{\boldsymbol{z}}_0$:

$$\hat{\boldsymbol{z}}_0^{\text{selected}} = \arg\min_{\hat{\boldsymbol{z}}_0^{(i)}} \|\hat{\boldsymbol{z}}_0^{(i)} - \bar{\boldsymbol{z}}_0\|_2^2. \tag{10}$$

This selected latent vector $\hat{\boldsymbol{z}}_0^{\text{selected}}$ is then transformed by the decoder $\mathcal{D}$ into the final image $\hat{\mathbf{x}}_0$, as represented by the equation $\hat{\mathbf{x}}_0 = \mathcal{D}(\hat{\boldsymbol{z}}_0^{\text{selected}})$.

## 4 Experiments

In this section, we present experiments to numerically evaluate our proposed method. First, we utilize a 2D toy example (Sec. 4.1) to visually illustrate the impact of our proposed Kernel Density Steering on the diffusion model sampling process, providing intuition for its mechanism. Subsequently, we demonstrate the effectiveness and practical applicability of our approach on two challenging real-world tasks: image super-resolution (SR) (Sec. 4.2.1) and image inpainting (Sec. 4.2.2).

### 4.1 KDS Sampling in a 2D Example Case

To visualize KDS's effect, we consider sampling from a 2D Mixture of Gaussians (MoG) target distribution $p(\boldsymbol{x}_0) = \sum_{c=1}^C \pi_c \mathcal{N}(\boldsymbol{x}_0; \boldsymbol{\mu}_c, \boldsymbol{\Sigma}_c)$. In this controlled setting, the exact score $\nabla_{\boldsymbol{x}_t} \log p_t(\boldsymbol{x}_t)$ of the noisy distribution is known. KDS is applied by adding its steering term (derived from Eq. 9 using the 2D particle states $\{\boldsymbol{x}_t^{(i)}\}_{i=1}^N$) to the update driven by the exact score within a DDIM sampler. As the data dimension here is very low, we let the patch size equal to the full data size.

As illustrated in Figure 2, KDS significantly sharpens the resulting sample distribution. Samples cluster more tightly around the true mode means ($\boldsymbol{\mu}_c$), effectively reducing the spread (variance) of samples within each mode compared to using the exact score alone. This occurs because KDS actively guides particles towards their respective mode centers—which correspond to regions of high sample density in the ensemble—thereby improving concentration of samples around each mode.

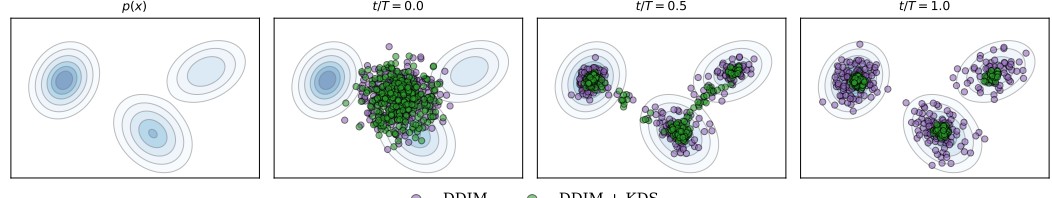

Figure 2: Kernel Density Steering (KDS) sharpens sample distributions in a 2D Mixture of Gaussians (MoG) toy problem. The target distribution $p(\boldsymbol{x}_0)$ consists of three distinct Gaussian modes. Samples are drawn using DDIM with the exact score function (purple dots) versus DDIM augmented with KDS (green dots, $N = 50$ particles). KDS guides particles through the reverse diffusion progresses, leading to significantly higher sample concentration at the mode peaks compared to standard DDIM.

Table 1: Super-Resolution Performance on DIV2K dataset with DDIM (50 steps)

| Method | PSNR | SSIM | LPIPS ($\downarrow$) | NIMA | DISTS ($\downarrow$) | MANIQA | CLIPIQA | FID ($\downarrow$) |
|---|---|---|---|---|---|---|---|---|
| LDM-SR | 22.05 | 0.531 | 0.307 | 4.922 | 0.179 | 0.390 | 0.594 | 20.89 |
| + KDS ($N = 10$) | **22.37**⋆ | **0.549**⋆ | **0.292**⋆ | **4.949**⋆ | **0.176**⋆ | **0.399**⋆ | **0.601**⋆ | **20.78**⋆ |
| DiffBIR | 21.56 | 0.488 | 0.377 | 5.195 | 0.223 | 0.566 | 0.730 | 32.28 |
| + KDS ($N = 10$) | **22.44**⋆ | **0.535**⋆ | **0.348**⋆ | **5.219**⋆ | **0.220**⋆ | **0.571**⋆ | **0.744**⋆ | **30.67**⋆ |
| SeeSR | 22.43 | 0.573 | 0.340 | 4.902 | 0.200 | 0.423 | 0.605 | 25.96 |
| + KDS ($N = 10$) | **22.79**⋆ | **0.587**⋆ | **0.313**⋆ | **5.026**⋆ | **0.191**⋆ | **0.488**⋆ | **0.679**⋆ | **25.44**⋆ |

Statistically significant differences ($P < 0.05$) compared with DDIM are marked with a ⋆.

## 4.2 Real-world Image Restoration

To demonstrate the effectiveness of our Kernel Density Steering (KDS), we compare its performance against baseline sampling method on image super-resolution and inpainting tasks. Our experiments utilize two widely used samplers (DDIM, DPM-Solver++). For each configuration, we contrast the results obtained with standard sampling versus KDS-enhanced sampling, ensuring fairness by using the same initial noise. We compare performance with and without KDS, using the same initial noise $\{z_T^{(i)}\}_{i=1}^N$ for fairness across an ensemble of $N = 10$ particles unless specified otherwise. Full experimental details, including hyperparameter settings (e.g., patch size, steering strength $\delta_t$) and comprehensive ablation studies, are provided in the Appendix.

Table 2: Super-Resolution Performance on real-world collected datasets with DDIM (50 steps)

| Datasets | RealSR | | | | | DrealSR | | | | |
|---|---|---|---|---|---|---|---|---|---|---|
| Method | PSNR | SSIM | LPIPS ($\downarrow$) | DISTS ($\downarrow$) | CLIPIQA | PSNR | SSIM | LPIPS ($\downarrow$) | DISTS ($\downarrow$) | CLIPIQA |
| LDM-SR | 24.01 | 0.666 | 0.308 | 0.211 | **0.624** | 26.13 | 0.689 | 0.342 | 0.222 | 0.610 |
| + KDS ($N = 10$) | **24.68**⋆ | **0.703**⋆ | **0.275**⋆ | **0.201**⋆ | 0.622 | **26.32**⋆ | **0.702**⋆ | **0.331**⋆ | **0.317**⋆ | **0.617**⋆ |
| DiffBIR | 23.21 | 0.610 | 0.370 | 0.250 | 0.689 | 23.99 | 0.551 | 0.491 | 0.293 | **0.701** |
| + KDS ($N = 10$) | **24.39**⋆ | **0.669**⋆ | **0.339**⋆ | **0.244**⋆ | **0.692**⋆ | **25.63** | **0.645** | **0.422**⋆ | **0.276**⋆ | 0.693⋆ |
| SeeSR | 24.29 | 0.710 | 0.279 | **0.204** | 0.588 | 26.97 | 0.750 | 0.300 | 0.218 | 0.625 |
| + KDS ($N = 10$) | **24.50**⋆ | **0.719**⋆ | **0.272**⋆ | 0.206⋆ | **0.640**⋆ | **27.42**⋆ | **0.765**⋆ | **0.287**⋆ | **0.212**⋆ | **0.651**⋆ |

Statistically significant differences ($P < 0.05$) compared with DDIM are marked with a ⋆.

### 4.2.1 Real-world Image Super-Resolution

We evaluated the performance of KDS for $4\times$ real-world super-resolution (SR). This evaluation utilized the LDM-SR [24], DiffBIR [4], and SeeSR [33] backbones across several datasets: DIV2K [56], RealSR [57], DrealSR [58]. Performance was evaluated using a comprehensive suite of metrics, including PSNR, SSIM [59], LPIPS [60], FID [61], NIMA [62], MANIQA [63], and CLIPIQA [64].

**Results:** Quantitative results consistently show KDS's benefits. Tables 1 and Tables 2 demonstrate that adding KDS significantly improves both distortion and perceptual metrics across different

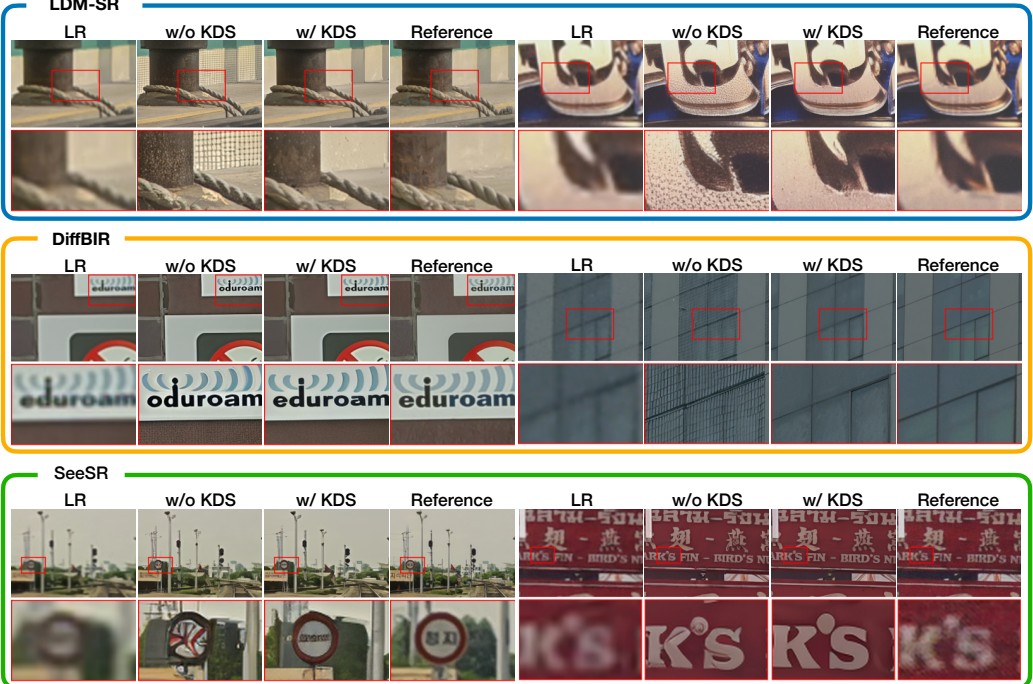

Figure 3: Qualitative comparison of $4\times$ Real-world Super-Resolution with KDS-enhanced DDIM sampling (Number of Particles $N = 10$). 'w/o KDS' shows results from baseline DDIM, while 'w/ KDS' shows results with KDS. KDS consistently produces images with improved sharpness, finer details, and reduced artifacts across LDM-SR, DiffBIR, and SeeSR backbones.

datasets, backbones, and samplers (DDIM, DPM-Solver++) compared to the respective baselines without KDS. Qualitative results in Figure 3 corroborate these findings, showing restorations with higher fidelity and fewer artifacts when using KDS.

**Compare with Best-of-N approaches:** While inference-time scaling allows for the generation of multiple candidate solutions, especially beneficial for low-quality inputs, selecting the optimal one from an N-particle ensemble is non-trivial. As Table 4 shows, using a non-reference metric like LIQE [65] to pick the best particle results in significantly lower performance compared to our kernel density steering (KDS) method. This demonstrates that, despite comparable computational costs, traditional post-sampling selection methods do not achieve the same level of performance as KDS.

Table 3: Inpainting Performance on ImageNet with LDM-inpainting backbone.

| Method | PSNR | SSIM | LPIPS($\downarrow$) | FID($\downarrow$) |
|---|---|---|---|---|
| DPM-Solver | 19.94 | 0.725 | 0.144 | 11.70 |
| + KDS ($N=10$) | **21.29** | **0.747** | **0.135** | **11.29** |
| DDIM | 21.03 | 0.736 | 0.140 | 11.47 |
| + KDS ($N=10$) | **21.35** | **0.748** | **0.131** | **11.18** |

Table 4: Comparison to Best-of-$N$ (BoN) using LIQE [65] on RealSR with DiffBIR.

| Method | $N$ | PSNR | LPIPS ($\downarrow$) | Time | Memory |
|---|---|---|---|---|---|
| DDIM | 1 | 23.21 | 0.370 | 7.9s | 8.0Gb |
| + BoN | 5 | 23.72 | 0.361 | 14.9s | 9.7Gb |
| + KDS | 5 | **24.17** | **0.349** | 15.6s | 9.7Gb |
| + BoN | 10 | 23.77 | 0.366 | 27.3s | 10.4Gb |
| + KDS | 10 | **24.39** | **0.339** | 28.4s | 10.4Gb |

### 4.2.2 Image Inpainting

We evaluate the performance of KDS (N=10) on a center box inpainting task using the ImageNet dataset. For this task, the central 30% square region of each image is masked. We employ a Latent Diffusion Model (LDM) for inpainting, initialized with weights from a pretrained text-to-image model that matches the architecture and size of Stable Diffusion v2 [24]. The LDM was fine-tuned on ImageNet dataset for random box inpainting. The training utilized a batch size of 1024 and a

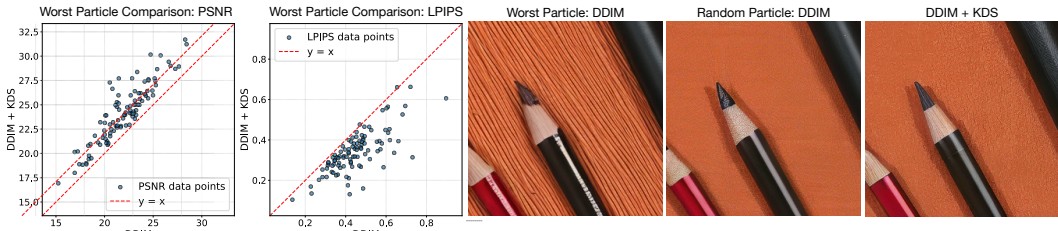

Figure 4: Robustness analysis of KDS on the RealSR dataset. **Left**: Scatter plots comparing the PSNR and LPIPS of the DDIM with KDS versus a worst-performing particle of standard DDIM ensemble ($N = 10$). KDS consistently improves the quality of the worst-case samples. **Right**: Qualitative examples comparing the worst-performing output from a DDIM ensemble with KDS-guided DDIM with the same random seed, demonstrating KDS's superior consistency and artifact reduction.

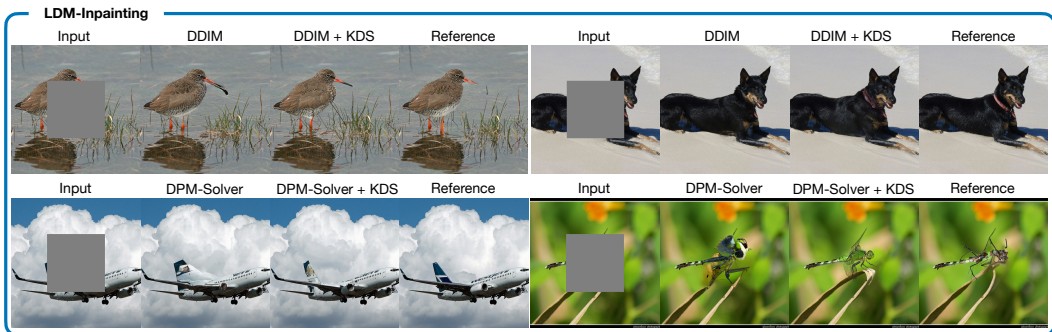

Figure 5: Box-inpainting performance of KDS. KDS generates more coherent and detailed inpainted regions compared to standard DDIM sampling.

learning rate of 1e-4, parameters empirically selected to maximize compute efficiency. Performance was measured using PSNR, SSIM, LPIPS, and FID.

As shown in Table 3, our proposed KDS improves quantitative inpainting performance. Furthermore, Figure 5 demonstrates that KDS generates inpainted regions with enhanced fidelity, greater coherence with surrounding image content, and more plausible details compared to standard sampling.

### 4.3 Analysis of KDS Robustness

A key objective of KDS is to enhance the reliability and consistency of diffusion-based restorations, particularly by reducing artifacts and improving fidelity.

Figure 4 demonstrates this improved stability. The scatter plots (left) compare the performance (PSNR and LPIPS) of DDIM enhanced by KDS against the worst-performing particle from a baseline DDIM ensemble (also $N = 10$) on RealSR dataset. Points above (PSNR) or below (LPIPS) the $y = x$ line signify that KDS improves the worst-case performance. These two scatter plots clearly demonstrate enhanced restoration reliability. Qualitative examples (right part of Figure 4) further illustrate KDS's consistency and artifact reduction.

### 4.4 Impact of KDS Hyperparameters: Number of Particles and Bandwidth

The performance of KDS, as an ensemble-based technique, is inherently linked to its key hyper-parameters: the number of particles $N$ and the kernel bandwidth $h$. We analyze their impact on SR performance (PSNR, SSIM, LPIPS, FID) using the LDM-SR model on the DIV2K dataset, as illustrated in Figure 6.

**Number of Particles $N$:** Increasing the ensemble size $N$ consistently enhances restoration quality. As shown in Figure 6, both distortion metrics (e.g., PSNR, SSIM) and perceptual metrics (e.g., LPIPS, FID) benefit, particularly as $N$ increases from small (e.g., $N \approx 5 - 10$), where kernel density estimates are less robust, to moderate (e.g., $N \approx 15 - 20$) values. While further increasing $N$ (e.g.,

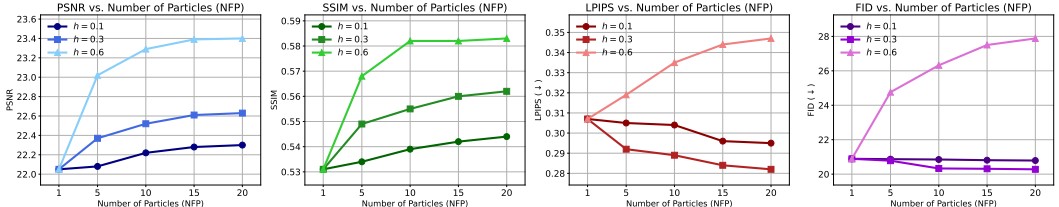

Figure 6: Influence of Particle Number ($N$) and Bandwidth ($h$) on Real-world SR. Performance metrics (PSNR, SSIM, LPIPS, FID) are plotted against the number of particles ($N$) for different bandwidth ($h$) settings on the DIV2K dataset, using LDM-SR backbone with KDS.

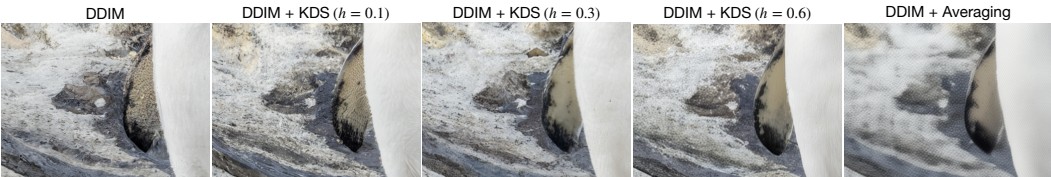

Figure 7: Influence of Bandwidth ($h$) on Real-world SR based on LDM-SR backbone. Note, DDIM + averaging means a post-sampling averaging across the whole ensemble.

beyond $N > 15$ in depicted scenarios) can yield additional improvements, the gains may become marginal, leading to saturation. This underscores the inherent trade-off between the computational cost, which scales linearly with $N$, and the achievable performance boost.

**Kernel Bandwidth $h$**: The kernel bandwidth $h$ plays a crucial role in balancing the perception-distortion trade-off. Figure 6 (different colored lines representing different $h$ settings) demonstrates that while larger ensembles generally improve performance, the choice of $h$ is critical. Excessively large bandwidths (e.g., $h = 0.6$ in some cases) can lead to over-smoothing, potentially enhancing distortion metrics like PSNR & SSIM at the cost of perceptual quality (higher LPIPS & FID). Conversely, a very small $h$ might not capture sufficient local consensus. Optimal $h$ selection is therefore essential for achieving the desired balance between quantitative accuracy and perceptual sharpness. Our experiments suggest that a moderate $h$ often provides a good compromise. Furthermore, in Appendix A.2, we show that standard automatic bandwidth selection strategies (e.g., median heuristic) achieve performance nearly identical to our manually tuned $h$. This demonstrates that KDS can be deployed effectively without tedious manual tuning, enhancing its practical applicability.

Further parameter analyses in Appendix A.2 confirm KDS is robust to hyperparameter choices, including different kernel functions (e.g., Gaussian, Laplacian). We also explore the trade-off between particle count ($N$) and patch size. As shown in the appendix, increasing $N$ enables the use of larger patch sizes, which enhances performance by better mitigating the curse of dimensionality.

## 5  Discussion

We further analyze KDS by comparing it to other inference-time techniques. We compare it against other mode-seeking sampling strategies (HDS [66], DGS [67]) and separately against other methods, like InDI [29], which are also designed to improve the perception-distortion trade-off.

**Comparison with HDS [66] and DGS [67].** To position KDS relative to other inference-time mode-seeking techniques, we compare it against High Density Sampling (HDS) [66] and Density-Guided Sampling (DGS) [67]. We implemented both methods with the DiffBIR [4] backbone and swept their key hyperparameters (time $t$ for HDS, quantile $q$ for DGS) to find their best-performing configurations on the RealSR dataset, with a full parameter sweep available in Appendix A.2.

As shown in Table 5, HDS achieves high PSNR at a severe cost to perceptual quality (LPIPS, CLIPIQA), while DGS offers moderate improvement. In contrast, KDS ($N = 10$) provides the best balance, uniquely achieving both a strong PSNR and the best perceptual scores (LPIPS, DISTS, CLIPIQA), confirming its superior perception-distortion trade-off.

The core conceptual difference lies in their assumptions. Both HDS [66] and DGS [67] are mode-seeking but rely on different assumptions compared with KDS. HDS assumes a unimodal Gaussian distribution, which is often violated by highly multi-modal IR posteriors. DGS, while non-parametric in a different sense, guides a single sample by progressively sharpening the marginal density $p_t(x_t)$ at each step, forcing it toward a single high-density mode. In sharp contrast, KDS makes no explicit assumption about the posterior's shape. It uses a non-parametric KDE on an ensemble of outputs ($\hat{z}_{0|t}$), making it better suited to finding a consensus within the multi-modal posteriors common in IR tasks.

**Comparison with InDI [29].** We also distinguish KDS from InDI [29], which also adjusts the distortion-perception trade-off at inference time. Their core approaches differ: InDI is not a posterior mode-seeking algorithm. It performs progressive refinement to the final estimation $z_0$, which does not aim to sample from the posterior mode. KDS, in contrast, directly seeks a consensus mode from the final clean output ensemble ($\hat{z}_0$). Moreover, KDS is a training-free, plug-and-play module, whereas InDI requires an integrated training and sampling pipeline.

Table 5: Comparison with other Mode-Seeking Methods on RealSR dataset with DiffBIR backbone.

| Method | PSNR $\pm\sigma$ | SSIM $\pm\sigma$ | LPIPS ($\downarrow$) $\pm\sigma$ | DISTS ($\downarrow$) $\pm\sigma$ | CLIPIQA $\pm\sigma$ |
|---|---|---|---|---|---|
| DiffBIR | 23.21±0.293 | 0.610±0.015 | 0.370±0.011 | 0.250±0.005 | 0.689±0.013 |
| + HDS [66] | 24.56±0.344⋆ | 0.646±0.020⋆ | 0.386±0.017⋆ | 0.274±0.016⋆ | 0.647±0.025⋆ |
| + DGS [67] | 24.05±0.287⋆ | 0.651±0.014⋆ | 0.355±0.012⋆ | 0.246±0.004⋆ | 0.687±0.013 |
| + KDS (ours) | 24.39±0.271⋆ | 0.669±0.013⋆ | 0.339±0.011⋆ | 0.244±0.004⋆ | 0.692±0.011⋆ |

Statistically significant differences ($P < 0.05$) compared with DDIM are marked with a ⋆.

## 6  Limitations & Future Work

**Limitations.** The primary limitation of KDS is its computational overhead, which scales linearly with particle count $N$. Although $N = 10 - 15$ provides a good performance-cost trade-off, this overhead could be mitigated by future work on adaptive ensemble sizes or more efficient KDE techniques.

**Future Work.** As a training-free, plug-and-play framework, KDS is highly generalizable. Promising directions include applying it to other inverse problems (e.g., deblurring, denoising, medical imaging) and video restoration, where its patch-wise mechanism could be extended to 3D spatio-temporal patches to enhance temporal consistency.

## 7  Conclusion

This work introduces Kernel Density Steering (KDS), a novel inference-time framework that enhances the fidelity and robustness of diffusion-based image restoration. The core of KDS lies in its local mode seeking strategy: it utilizes an $N$-particle ensemble and computes patch-wise kernel density estimation (KDE) gradients from predicted clean latent patches. These gradients guide samples towards shared, high-density regions, effectively steering them away from spurious modes to produce robust, high-fidelity restorations. As a plug-and-play approach, KDS requires no retraining, external verifiers, or explicit degradation models, making it broadly applicable.

## Acknowledgements

The authors would like to thank our colleagues Keren Ye, Viraj Shah, Ashwini Pokle and Mo Zhou for reviewing the manuscript and providing valuable feedback.

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

# A    Supplementary Material

This supplementary material provides further details to the main paper. It covers the following aspects:

- **Section A.1: Experiment Details for Real-world Image Super-Resolution**: Including dataset descriptions, backbone model configurations, an in-depth discussion of the patch-wise KDS mechanism with ablation studies, interaction of KDS with various diffusion samplers (DDIM, DPM-Solver++), and additional comparative results on severe degradation datasets.

- **Section A.3: Experiment Details for Image Inpainting**: Covering the datasets used and the hyperparameter settings for the inpainting task.

- **Section A.4: Additional Visual Results**: Presenting more qualitative examples for both super-resolution and image inpainting tasks to further demonstrate the efficacy of KDS.

## A.1    Experiment details: Real-world Image Super Resolution

### A.1.1    Datasets

Our real-world image super-resolution experiments utilized a synthesized dataset derived from DIV2K [56], alongside two real-world datasets: RealSR [57] and DRealSR [58]. For the DIV2K-based synthesized data, we used the same dataset provided by StableSR [11], which consists of 3,000 randomly cropped patches (resolution: $512 \times 512$ each) from the DIV2K validation set [56]. Subsequently, we generated corresponding low-resolution (LR) images (resolution: $128 \times 128$)) using the degradation model adopted in Real-ESRGAN [68]. For the RealSR [57] and DRealSR [58] datasets, we adhered to the protocol from [11] to center-crop the provided LR images to $128 \times 128$.

### A.1.2    Backbone Model Configurations

To assess the effectiveness of our proposed KDS method, we applied it to three established backbone models: LDM-SR [24], DiffBIR [4], and SeeSR [33]. For LDM-SR, the LR image was directly employed as a conditional input, concatenated with the LDM's primary input. For DiffBIR [4] and SeeSR [33], we adopted the hyperparameter settings recommended on their official GitHub pages. This included their specified parameters for text prompting, classifier-free guidance, and the use of their official pre-trained model checkpoints.

### A.1.3    Patch-wise KDS Mechanism: Configuration and Discussion

As introduced in Section 3.1, we employ a patch-wise mechanism to facilitate mode-seeking within the high-dimensional latent space. The motivation is that the accuracy of the mode-seeking process is intrinsically linked to both the number of samples available for kernel density estimation (KDE) and the dimensionality of these samples. In our case, direct mode-seeking on the entire $64 \times 64 \times 8$ latent vector $z$ (which encodes features for the $512 \times 512 \times 3$ image space) is impractical with a practical number of particles (e.g., 10-20).

**Patch Size Configuration:** To effectively implement our patch-wise strategy, the patch size is chosen to balance accurate mode-seeking with computational efficiency. Accurate mode-seeking across the entire latent space would necessitate thousands of samples, which is impractical for real-world applications. Therefore, to achieve robust estimation with minimal samples, we set the patch size to $1 \times 1$. This processes each spatial location $(h, w)$ in the latent map as an individual 8-dimensional vector (i.e., batches of $1 \times 1 \times 8$ vectors). This strategic choice significantly reduces the dimensionality for each kernel density estimation (KDE), enabling accurate estimation with limited samples while leveraging the rich, learned features of the 8 channels. Consequently, this facilitates more effective mode-seeking with a limited particle count. Ablation studies (Table 6) demonstrate that with a restricted particle count ($N = 10$), larger patch sizes lead to inaccurate mode-seeking and sub-optimal performance.

**Steering Strength $\delta_t$ Configuration:** Another key hyperparameter in KDS is the steering strength, denoted as $\delta_t$. During the diffusion process, the sampling procedure exhibits varying sensitivity to guidance. Specifically, at later stages of the sampling process (i.e., smaller $t$ values, approaching

Table 6: Ablation Study on Patch-size ($N = 10$)

|  | PSNR | SSIM | LPIPS ↓ | FID |
|---|---|---|---|---|
| DDIM | 22.05 | 0.531 | 0.307 | 20.89 |
| + KDS (patch-size=1) | 22.52 | 0.555 | 0.289 | 20.33 |
| + KDS (patch-size=4) | 22.35 | 0.547 | 0.296 | 20.77 |
| + KDS (patch-size=16) | 22.17 | 0.538 | 0.304 | 20.86 |

the data), the model can be more sensitive. To ensure stability and effective guidance, we define $\delta_t$ conditionally based on the timestep $t$. Assuming $T$ is the total number of diffusion steps, we set:

$$\delta_t = \begin{cases} 0 & \text{if } t/T < 0.3 \\ 0.3 & \text{if } t/T \geq 0.3 \end{cases} \tag{11}$$

Note that, this hyperparameter setting is fixed for all the experiments across different applications and backbones.

#### A.1.4 Integration with Standard Diffusion Samplers

As discussed in the main paper, KDS is a flexible, plug and play approach that can be applied to all existing samplers. In this section, we will detailed introduce how to apply KDS to DDIM, and DPM-solver and how to interactive with Classifier-Free Guidance.

**Interaction with DDIM:** To better illustrate the plug-and-play nature of KDS, we provide pseudocode for three scenarios:

1. **Algorithm 1:** Standard DDIM sampling.
2. **Algorithm 2:** DDIM integrated with KDS.
3. **Algorithm 3:** DDIM combined with Classifier-Free Guidance (CFG) and KDS.

As demonstrated in the pseudocode, KDS functions as a straightforward plug-in module (Line 5 - 15 in Algorithm 2). It enhances the predicted ensemble $\hat{\mathbf{Z}}_{0|t}^{\text{pred}}$ and maintain the rest sampling design of the base sampler.

**Interaction with Classifier-Free Guidance (CFG):** CFG is frequently used in conditional diffusion models for SR (like DiffBIR [4], SeeSR [33]) to further improve perceptual quality. CFG is applied by adjusting the noise prediction, commonly via the extrapolation formula

$$\tilde{\boldsymbol{\epsilon}}(\boldsymbol{z}_t, t, \boldsymbol{c}) = \boldsymbol{\epsilon}(\boldsymbol{z}_t, t, \varnothing) + w(\boldsymbol{\epsilon}(\boldsymbol{z}_t, t, \boldsymbol{c}) - \boldsymbol{\epsilon}(\boldsymbol{z}_t, t, \varnothing)), \tag{12}$$

where $w$ is the guidance scale. While higher $w$ can enhance perception, it sometimes introduces artifacts. We investigated how KDS interacts with CFG by varying $w$ in the DiffBIR model using DDIM sampling on the DrealSR dataset. As shown in Table 7, KDS consistently boosts performance across different CFG strengths ($w = 1, 2, 4$). For each value of $w$, adding KDS leads to substantial improvements in PSNR and SSIM, along with generally better perceptual metrics (LPIPS, NIMA, CLIPIQA). This suggests that KDS provides benefits complementary to CFG, enhancing fidelity without hindering the perceptual adjustments offered by CFG.

Table 7: Performance of DDIM with Varying CFG Weights $w$ on DrealSR Dataset.

| Method | PSNR | SSIM | LPIPS ($\downarrow$) | DISTS($\downarrow$) |
|---|---|---|---|---|
| DDIM ($w = 1$) | 25.11 | 0.576 | 0.492 | 0.298 |
| + KDS ($N = 10$) | **26.94** | **0.677** | **0.427** | **0.283** |
| DDIM ($w = 2$) | 24.75 | 0.569 | 0.486 | 0.293 |
| + KDS ($N = 10$) | **26.60** | **0.667** | **0.428** | **0.278** |
| DDIM ($w = 4$) | 23.99 | 0.551 | 0.491 | 0.293 |
| + KDS ($N = 10$) | **25.63** | **0.645** | **0.422** | **0.276** |
| DDIM ($w = 6$) | 23.27 | 0.534 | 0.497 | 0.296 |
| + KDS ($N = 10$) | **25.04** | **0.629** | **0.440** | **0.282** |

---

**Algorithm 1** Standard DDIM Sampling

---

**Require:** Model $\epsilon_\theta$, condition: $\boldsymbol{c}$, Schedule $\bar{\alpha}_t$
1: $\boldsymbol{z}_T \sim \mathcal{N}(0, \mathbf{I})$ ▷ Init noise
2: **for** $t = T, \ldots, 1$ **do**
3: $\quad \boldsymbol{\epsilon}_{\theta,t} \leftarrow \boldsymbol{\epsilon}_\theta(\boldsymbol{z}_t, t, \boldsymbol{c})$ ▷ Predict noise
4: $\quad \hat{\boldsymbol{z}}_{0|t} \leftarrow (\boldsymbol{z}_t - \sqrt{1 - \bar{\alpha}_t}\boldsymbol{\epsilon}_{\theta,t})/\sqrt{\bar{\alpha}_t}$ ▷ Predict $\boldsymbol{z}_0$
5: $\quad \boldsymbol{\epsilon}'_t \leftarrow (\boldsymbol{z}_t - \sqrt{\bar{\alpha}_t}\hat{\boldsymbol{z}}_{0|t})/\sqrt{1 - \bar{\alpha}_t}$ ▷ Update direction based on $\hat{\boldsymbol{z}}_{0|t}$
6: $\quad \boldsymbol{z}_{t-1} \leftarrow \sqrt{\bar{\alpha}_{t-1}}\hat{\boldsymbol{z}}_{0|t} + \sqrt{1 - \bar{\alpha}_{t-1}}\boldsymbol{\epsilon}'_t$ ▷ DDIM step
7: **end for**
8: **return** $\boldsymbol{z}_0$

---

---

**Algorithm 2** DDIM + KDS

---

**Require:** Model $\boldsymbol{\epsilon}_\theta$, Schedule $\bar{\alpha}_t$, condition: $\boldsymbol{c}$, Number of particles: $N$, bandwidth: $h$, steering
  strength: $\delta_t$, PatchSize
1: $\boldsymbol{Z}_T \sim \mathcal{N}(0, \mathbf{I})$ (ensemble of $N$ samples)          ▷ Init noise ensemble
2: **for** $t = T, \ldots, 1$ **do**
3:  $\mathbf{E}_{\theta,t} \leftarrow \boldsymbol{\epsilon}_\theta(\boldsymbol{Z}_t, t, \boldsymbol{c})$             ▷ Predict ensemble noise
4:  $\hat{\boldsymbol{Z}}_{0|t}^{\text{pred}} \leftarrow (\boldsymbol{Z}_t - \sqrt{1 - \bar{\alpha}_t}\mathbf{E}_{\theta,t})/\sqrt{\bar{\alpha}_t}$       ▷ Predict $\mathbf{x}_0$ ensemble
5:  **Patches** $\leftarrow$ Patchify($\hat{\boldsymbol{Z}}_{0|t}^{\text{pred}}$)    ▷ Extract all non-overlapped patches: **Patches**$[k, loc]$.
6:  **for** each patch location $j$ **do**   ▷ This loop over patch locations, can be executed **in parallel**.
7:   $\boldsymbol{P}_j \leftarrow$ **Patches**$[:, j]$         ▷ Ensemble of $N$ original patches at location $j$.
8:   **for** $i = 1, \ldots, N$ **do**   ▷ For particle $i$'s patch at location $j$, can be computed **in parallel**.
9:    $\boldsymbol{p}_j^{(i)} \leftarrow \boldsymbol{P}_j[i]$          ▷ Patch from particle $i$ at location $j$.

10:    $\mathbf{m}(\boldsymbol{p}_j^{(i)}) \leftarrow \dfrac{\sum_{k=1}^N G\left(\frac{\|\boldsymbol{p}_j^{(i)} - \boldsymbol{P}_j^{(k)}\|^2}{h^2}\right)\boldsymbol{P}_j^{(k)}}{\sum_{k=1}^N G\left(\frac{\|\boldsymbol{p}_j^{(i)} - \boldsymbol{P}_j^{(k)}\|^2}{h^2}\right)} - \boldsymbol{p}_j^{(i)}$     ▷ Mean shift vector (Eq. 8).

11:    $\hat{\boldsymbol{p}}_j^{(i),\text{KDS}} \leftarrow \boldsymbol{p}_j^{(i)} + \delta_t\mathbf{m}(\boldsymbol{p}_j^{(i)})$       ▷ Apply steering (Eq. 9)
12:    **Patches**$[i, j] \leftarrow \hat{\boldsymbol{p}}_j^{(i),\text{KDS}}$     ▷ Update the patch set with the guided patch.
13:   **end for**
14:  **end for**
15:  $\hat{\boldsymbol{Z}}_{0|t}^{\text{KDS}} \leftarrow$ Unpatchify(**Patches**)      ▷ Reconstruct guided latent prediction.
16:  $\mathbf{E}_t' \leftarrow (\boldsymbol{Z}_t - \sqrt{\bar{\alpha}_t}\hat{\boldsymbol{Z}}_{0|t}^{\text{KDS}})/\sqrt{1 - \bar{\alpha}_t}$       ▷ Update direction
17:  $\boldsymbol{Z}_{t-1} \leftarrow \sqrt{\bar{\alpha}_{t-1}}\hat{\boldsymbol{Z}}_{0|t}^{\text{KDS}} + \sqrt{1 - \bar{\alpha}_{t-1}}\mathbf{E}_t'$      ▷ DDIM step
18: **end for**
19: **return** $\hat{\boldsymbol{Z}}_0^{\text{KDS}}$            ▷ Return KDS-guided result

---

---

**Algorithm 3** DDIM + CFG + KDS

---

**Require:** Model $\boldsymbol{\epsilon}_\theta$, Schedule $\bar{\alpha}_t$, condition: $\boldsymbol{c}$, Number of particles: $N$, bandwidth: $h$, steering
  strength: $\delta_t$, CFG strength: $w$, PatchSize
1: $\boldsymbol{Z}_T \sim \mathcal{N}(0, \mathbf{I})$ (ensemble of $N$ samples)          ▷ Init noise ensemble
2: **for** $t = T, \ldots, 1$ **do**
3:  $\mathbf{E}_{\theta,t} \leftarrow \boldsymbol{\epsilon}_\theta(\boldsymbol{Z}_t, t, \varnothing) + w(\boldsymbol{\epsilon}_\theta(\boldsymbol{Z}_t, t, \boldsymbol{c}) - \boldsymbol{\epsilon}_\theta(\boldsymbol{Z}_t, t, \varnothing))$   ▷ Predict ensemble noise with CFG
4:  $\hat{\boldsymbol{Z}}_{0|t}^{\text{pred}} \leftarrow (\boldsymbol{Z}_t - \sqrt{1 - \bar{\alpha}_t}\mathbf{E}_{\theta,t})/\sqrt{\bar{\alpha}_t}$       ▷ Predict $\mathbf{x}_0$ ensemble
5:  $\hat{\boldsymbol{Z}}_{0|t}^{\text{KDS}} \leftarrow$ **Patch-wise KDS**($\hat{\boldsymbol{Z}}_{0|t}^{\text{pred}}$)     ▷ Same as Step 5-15 in Algorithm 2
6:  $\mathbf{E}_t' \leftarrow (\boldsymbol{Z}_t - \sqrt{\bar{\alpha}_t}\hat{\boldsymbol{Z}}_{0|t}^{\text{KDS}})/\sqrt{1 - \bar{\alpha}_t}$     ▷ Update direction based on KDS-guided $\mathbf{x}_0$
7:  $\boldsymbol{Z}_{t-1} \leftarrow \sqrt{\bar{\alpha}_{t-1}}\hat{\boldsymbol{Z}}_{0|t}^{\text{KDS}} + \sqrt{1 - \bar{\alpha}_{t-1}}\mathbf{E}_t'$     ▷ DDIM step with KDS-guided $\mathbf{x}_0$
8: **end for**
9: **return** $\hat{\boldsymbol{Z}}_0^{\text{KDS}}$            ▷ Return KDS-guided result

---

---

**Algorithm 4** DPM-Solver++.

---

**Require:** initial value $\boldsymbol{Z}_T$, time steps $\{t_i\}_{i=0}^M$ and $\{s_i\}_{i=1}^M$, data prediction model $\boldsymbol{Z}_\theta$.
1: $\boldsymbol{Z}_T \sim \mathcal{N}(0, \mathbf{I})$ (ensemble of $N$ samples)  $\qquad\qquad\qquad\qquad$ ▷ Init noise ensemble
2: $\tilde{\boldsymbol{Z}}_{t_0} \leftarrow \boldsymbol{Z}_T$.
3: **for** $i \leftarrow 1$ to $M$ **do**
4: $\qquad h_i \leftarrow \lambda_{t_i} - \lambda_{t_{i-1}}$
5: $\qquad r_i \leftarrow \frac{\lambda_{s_i} - \lambda_{t_{i-1}}}{h_i}$
6: $\qquad \hat{\boldsymbol{Z}}_{0|t}^{\text{pred}} \leftarrow \boldsymbol{Z}_\theta(\tilde{\boldsymbol{Z}}_{t_{i-1}}, t_{i-1})$
7: $\qquad \boldsymbol{U}_i \leftarrow \frac{\sigma_{s_i}}{\sigma_{t_{i-1}}} \tilde{\boldsymbol{Z}}_{t_{i-1}} - \alpha_{s_i} \left( e^{-r_i h_i} - 1 \right) \hat{\boldsymbol{Z}}_{0|t}^{\text{pred}}$
8: $\qquad \hat{\boldsymbol{U}}_{0|s}^{\text{pred}} \leftarrow \boldsymbol{Z}_\theta(\boldsymbol{U}_i, s_i)$
9: $\qquad \boldsymbol{D}_i \leftarrow (1 - \frac{1}{2r_i}) \hat{\boldsymbol{Z}}_{0|t}^{\text{pred}} + \frac{1}{2r_i} \hat{\boldsymbol{U}}_{0|s}^{\text{pred}}$
10: $\qquad \tilde{\boldsymbol{Z}}_{t_i} \leftarrow \frac{\sigma_{t_i}}{\sigma_{t_{i-1}}} \tilde{\boldsymbol{Z}}_{t_{i-1}} - \alpha_{t_i} \left( e^{-h_i} - 1 \right) \boldsymbol{D}_i$
11: **end for**
12: **return** $\tilde{\boldsymbol{Z}}_{t_M}$

---

---

**Algorithm 5** DPM-Solver++ with KDS.

---

**Require:** initial value $\boldsymbol{Z}_T$, time steps $\{t_i\}_{i=0}^M$ and $\{s_i\}_{i=1}^M$, data prediction model $\boldsymbol{Z}_\theta$.
1: $\boldsymbol{Z}_T \sim \mathcal{N}(0, \mathbf{I})$ (ensemble of $N$ samples)  $\qquad\qquad\qquad\qquad$ ▷ Init noise ensemble
2: $\tilde{\boldsymbol{Z}}_{t_0} \leftarrow \boldsymbol{Z}_T$.
3: **for** $i \leftarrow 1$ to $M$ **do**
4: $\qquad h_i \leftarrow \lambda_{t_i} - \lambda_{t_{i-1}}$
5: $\qquad r_i \leftarrow \frac{\lambda_{s_i} - \lambda_{t_{i-1}}}{h_i}$
6: $\qquad \hat{\boldsymbol{Z}}_{0|t}^{\text{pred}} \leftarrow \boldsymbol{Z}_\theta(\tilde{\boldsymbol{Z}}_{t_{i-1}}, t_{i-1})$
7: $\qquad \hat{\boldsymbol{Z}}_{0|t}^{\text{KDS}} \leftarrow \textbf{Patch-wise KDS}(\hat{\boldsymbol{Z}}_{0|t}^{\text{pred}})$  $\qquad\qquad$ ▷ Same as Step 5-15 in Algorithm 2
8: $\qquad \boldsymbol{U}_i \leftarrow \frac{\sigma_{s_i}}{\sigma_{t_{i-1}}} \tilde{\boldsymbol{Z}}_{t_{i-1}} - \alpha_{s_i} \left( e^{-r_i h_i} - 1 \right) \hat{\boldsymbol{Z}}_{0|t}^{\text{KDS}}$
9: $\qquad \hat{\boldsymbol{U}}_{0|s}^{\text{pred}} \leftarrow \boldsymbol{Z}_\theta(\boldsymbol{U}_i, s_i)$
10: $\qquad \hat{\boldsymbol{U}}_{0|s}^{\text{KDS}} \leftarrow \textbf{Patch-wise KDS}(\hat{\boldsymbol{U}}_{0|s}^{\text{pred}})$  $\qquad\qquad$ ▷ Same as Step 5-15 in Algorithm 2
11: $\qquad \boldsymbol{D}_i \leftarrow (1 - \frac{1}{2r_i}) \hat{\boldsymbol{Z}}_{0|t}^{\text{KDS}} + \frac{1}{2r_i} \hat{\boldsymbol{U}}_{0|s}^{\text{KDS}}$
12: $\qquad \tilde{\boldsymbol{Z}}_{t_i} \leftarrow \frac{\sigma_{t_i}}{\sigma_{t_{i-1}}} \tilde{\boldsymbol{Z}}_{t_{i-1}} - \alpha_{t_i} \left( e^{-h_i} - 1 \right) \boldsymbol{D}_i$
13: **end for**
14: **return** $\tilde{\boldsymbol{Z}}_{t_M}$

---

### A.1.5 Additional Comparisons on Various Degradations and Samplers

In this subsection, we further evaluate KDS's performance on several additional real-world SR degradations and its effectiveness as a plug-in module for DPM-Solver++ [69]. We introduce a novel dataset, DF2K, generated by synthesizing 3,000 randomly degraded image pairs from the original DF2K dataset. While adopting the Real-ESRGAN pipeline, we employed hyperparameters that introduce more significant blur, noise, and JPEG artifacts, making it a more challenging benchmark compared to standard degradation levels, such as those in DIV2K. As shown in Table 8, KDS consistently improves performance on both the challenging DF2K dataset and the standard DIV2K degradation dataset.

Table 8: Performance with LDM-SR backbone on different Real-world SR degradation levels.

| Datasets | DF2k | | | | | DIV2k | | | | |
|---|---|---|---|---|---|---|---|---|---|---|
| Metrics | PSNR | SSIM | LPIPS ($\downarrow$) | NIMA | FID ($\downarrow$) | PSNR | SSIM | LPIPS ($\downarrow$) | NIMA | FID ($\downarrow$) |
| DPM-Solver++ | 23.11 | 0.579 | 0.276 | 4.968 | 18.76 | 22.06 | 0.532 | 0.306 | 4.922 | 20.88 |
| + KDS | **23.70** | **0.594** | **0.265** | **4.972** | **18.38** | **22.29** | **0.542** | **0.290** | **4.947** | **20.65** |
| DDIM | 22.88 | 0.542 | 0.276 | 4.930 | 18.59 | 22.05 | 0.531 | 0.307 | 4.922 | 20.89 |
| + KDS | **23.71** | **0.597** | **0.261** | **4.943** | **18.11** | **22.37** | **0.549** | **0.292** | **4.949** | **20.78** |

### A.1.6 Additional comparison with Best-of-N approaches:

While inference-time scaling allows for generating multiple candidate solutions, particularly useful for low-quality inputs, selecting the optimal one from an N-particle ensemble remains a challenge. In our main paper, we didn't cover the full scope of this experiment. Here, we expand on that by including more metrics. As Table 9 now demonstrates, using non-reference metrics like LIQE [65] or ClipiQA to pick the best particle from an N-selection (i.e., "best LIQE best of N" or "best CLIPIQA best of N") results in significantly lower performance compared to our Kernel Density Steering (KDS) method. This shows that, despite comparable computational costs, traditional post-sampling selection methods don't achieve the same performance level as KDS. As shown in Figure 8, KDS method achieves the most stable performance compared with both BoN baselines, which suffers from the artifacts which confused the non-reference metrics.

Table 9: Comparison of BoN Selection Methods

| Method | PSNR | SSIM | LPIPS ($\downarrow$) | DISTS ($\downarrow$) | LIQE | CLIPIQA | Time(s) | Memory |
|---|---|---|---|---|---|---|---|---|
| DDIM | 23.21 | 0.610 | 0.370 | 0.250 | 4.046 | 0.689 | 7.9s | 8.0Gb |
| + BoN (LIQE) | 23.72 | 0.622 | 0.361 | 0.246 | **4.351** | 0.741 | 27.3s | 10.4Gb |
| + BoN (CLIPIQA) | 23.01 | 0.592 | 0.382 | 0.247 | 4.187 | **0.774** | 28.9s | 10.4Gb |
| + KDS | **24.39** | **0.669** | **0.339** | **0.245** | 3.819 | 0.692 | 28.4s | 10.4Gb |

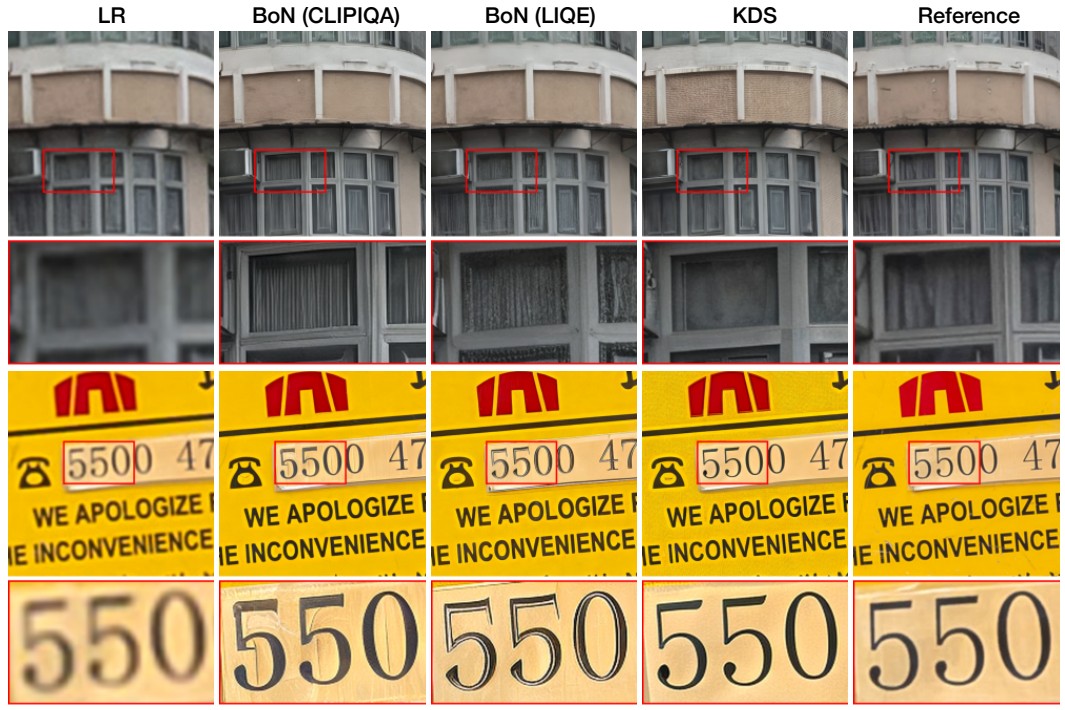

| LR | BoN (CLIPIQA) | BoN (LIQE) | KDS | Reference |

Figure 8: Visual comparison between KDS and Best-of-N (BoN) selection. BoN (CLIPIQA) and BoN (LIQE) means the best particle in terms of these two metrics correspondingly.

### A.1.7 KDS with post-hoc Best-of-N Selection:

We also investigate replacing the standard KDS final particle selection (Equation 10) with a post-hoc BoN selection. This involves first guiding the entire ensemble using KDS and then applying a non-reference metric (CLIPIQA) to select the best particle from the guided ensemble.

As shown in Table 10, this combination further validates our approach. KDS first enhances the overall quality of the ensemble, making the subsequent BoN selection more reliable. Without KDS, BoN selects from a noisy set of candidates and favors samples with artifacts that score well. With KDS, the candidates are already steered toward a high-fidelity consensus, providing a cleaner, more faithful set for the BoN selector. This results in a better perceptual–distortion trade-off than using BoN alone.

In summary, KDS provides a principled, task-aligned alternative to BoN by enforcing ensemble consistency. It improves performance across the board and can even strengthen BoN when the two are combined.

Table 10: Comparison of BoN and BoN + KDS Methods

| Method | PSNR | SSIM | LPIPS ($\downarrow$) | DISTS ($\downarrow$) | LIQE | CLIPIQA |
|---|---|---|---|---|---|---|
| DDIM | 23.21 | 0.610 | 0.370 | 0.250 | 4.046 | 0.689 |
| + BoN (CLIPIQA) | 23.01 | 0.592 | 0.382 | 0.247 | **4.187** | **0.774** |
| + KDS | **24.39** | **0.669** | **0.339** | **0.245** | 3.819 | 0.692 |
| + KDS+ BoN (CLIPIQA) | 23.69 | 0.636 | 0.359 | 0.243 | 4.161 | 0.770 |

## A.2  Additional Ablation Studies

This section provides the full empirical results for the ablation studies and comparisons mentioned in the main paper, including the full hyperparameter sweep for mode-seeking baselines and detailed ablations on KDS's components.

### A.2.1  Mode-Seeking Method Parameter Sweep

Table 11 provides the full parameter sweep for HDS [66] and DGS [67] on the RealSR dataset, supporting the analysis in Section 5.

Table 11: Full parameter sweep for HDS [66] and DGS [67] on RealSR.

| Method | PSNR | SSIM | LPIPS ($\downarrow$) | DISTS ($\downarrow$) | CLIPIQA |
|---|---|---|---|---|---|
| DiffBIR (Baseline) | 23.21 | 0.610 | 0.370 | 0.250 | 0.689 |
| + HDS (t=20) | 23.95 | 0.614 | 0.382 | 0.254 | 0.651 |
| + HDS (t=50) | 24.33 | 0.633 | 0.385 | 0.266 | 0.650 |
| + HDS (t=100) | 24.56 | 0.646 | 0.386 | 0.274 | 0.647 |
| + HDS (t=200) | 25.00 | 0.665 | 0.378 | 0.288 | 0.567 |
| + HDS (t=500) | 25.79 | 0.679 | 0.377 | 0.309 | 0.295 |
| + DGS (q=0.1) | 22.57 | 0.580 | 0.381 | 0.253 | 0.679 |
| + DGS (q=0.3) | 22.95 | 0.598 | 0.375 | 0.252 | 0.674 |
| + DGS (q=0.5) | 23.21 | 0.610 | 0.370 | 0.249 | 0.689 |
| + DGS (q=0.99) | 24.05 | 0.651 | 0.355 | 0.246 | 0.687 |
| + DGS (q=0.999) | 24.03 | 0.647 | 0.357 | 0.246 | 0.680 |
| + KDS (N=10) | **24.39** | **0.669** | **0.339** | **0.244** | **0.692** |

### A.2.2  Ablation on Guidance and Kernel Functions

Table 12 investigates the choice of kernel function and compares our mode-seeking approach to a simpler L2 guidance. The results show consistent performance between Gaussian and Laplacian kernels, demonstrating robustness. Both kernel-based methods outperform the L2 guidance baseline in perceptual metrics, validating our mode-seeking design.

Table 12: KDS with different guidance and kernels on RealSR dataset.

| Method | PSNR | SSIM | LPIPS ($\downarrow$) | DISTS ($\downarrow$) | CLIPIQA |
|---|---|---|---|---|---|
| DiffBIR | 23.21 | 0.610 | 0.370 | 0.250 | 0.689 |
| + KDS (Gaussian) | **24.39** | 0.669 | 0.339 | 0.244 | **0.692** |
| + KDS (Laplacian) | 24.30 | 0.665 | **0.337** | **0.241** | 0.690 |
| + KDS (L2) | **24.63** | **0.679** | 0.349 | 0.257 | 0.663 |

### A.2.3  Ablation on Automatic Bandwidth Selection

Table 13 demonstrates KDS's robustness to bandwidth selection. An automatic *median heuristic* strategy (auto bw) achieves performance nearly identical to the manually tuned bandwidth (bw=0.3), confirming KDS is not reliant on tedious tuning.

Table 13: Effect of automatic bandwidth selection for KDS.

| Method | PSNR | SSIM | LPIPS ($\downarrow$) | DISTS ($\downarrow$) | CLIPIQA |
|---|---|---|---|---|---|
| DiffBIR | 23.21 | 0.610 | 0.370 | 0.250 | 0.689 |
| + KDS (bw=0.3) | **24.39** | **0.669** | 0.339 | 0.244 | **0.692** |
| + KDS (auto bw) | 24.24 | 0.661 | **0.338** | **0.237** | 0.691 |

### A.2.4   Ablation on Patch Size and Particle Count

Table 14 explores the trade-off between patch size and particle count ($N$). As shown, increasing the number of particles (e.g., from $N = 10$ to $N = 40$) allows for the effective use of a larger patch size (from 1 to 4), which in turn enhances overall performance by better mitigating the curse of dimensionality.

Table 14: Ablation Study on patch size and particle count ($N$).

|  | PSNR | SSIM | LPIPS ↓ | FID ↓ |
|---|---|---|---|---|
| DDIM | 22.05 | 0.531 | 0.307 | 20.89 |
| + KDS (patch-size=1, $N = 10$) | 22.52 | 0.555 | 0.289 | 20.33 |
| + KDS (patch-size=4, $N = 40$) | 22.70 | 0.563 | 0.281 | 20.08 |

### A.3 Experiment details: Image Inpainting

**Datasets** We generated our inpainting test set by center cropping $30\%$ square region of each image. We generate used first 1,000 images from ImageNet testset.

**Hyperparameters:** Similar to real-world SR settings, we fixed the patch size to 1, bandwidth $h$ to 0.3, steering strength $\delta_t$ same as introduced in (Eq: 11).

### A.4 Additional Visual Results

This section provides additional qualitative results to visually demonstrate the effectiveness of Kernel Density Steering (KDS). The figures included are:

- **Figure 9, Figure 10 and Figure 11:** These figures showcase super-resolution performance on the DIV2K dataset using LDM-SR, DiffBIR and SeeSR backbones, respectively. They illustrate improvements in sharpness and detail recovery achieved with KDS.

- **Figure 12:** This figure highlights KDS's robustness, demonstrating its performance on the more challenging DF2K real-world SR dataset with the LDM-SR backbone.

- **Figures 13, 14, and 15:** These figures display image inpainting results on the ImageNet dataset using LDM-inpainting. Each figure presents all 10 particles sampled with standard DDIM versus DDIM enhanced with KDS. They visually confirm KDS's ability to improve fidelity and reduce artifacts across the ensemble for the inpainted regions.

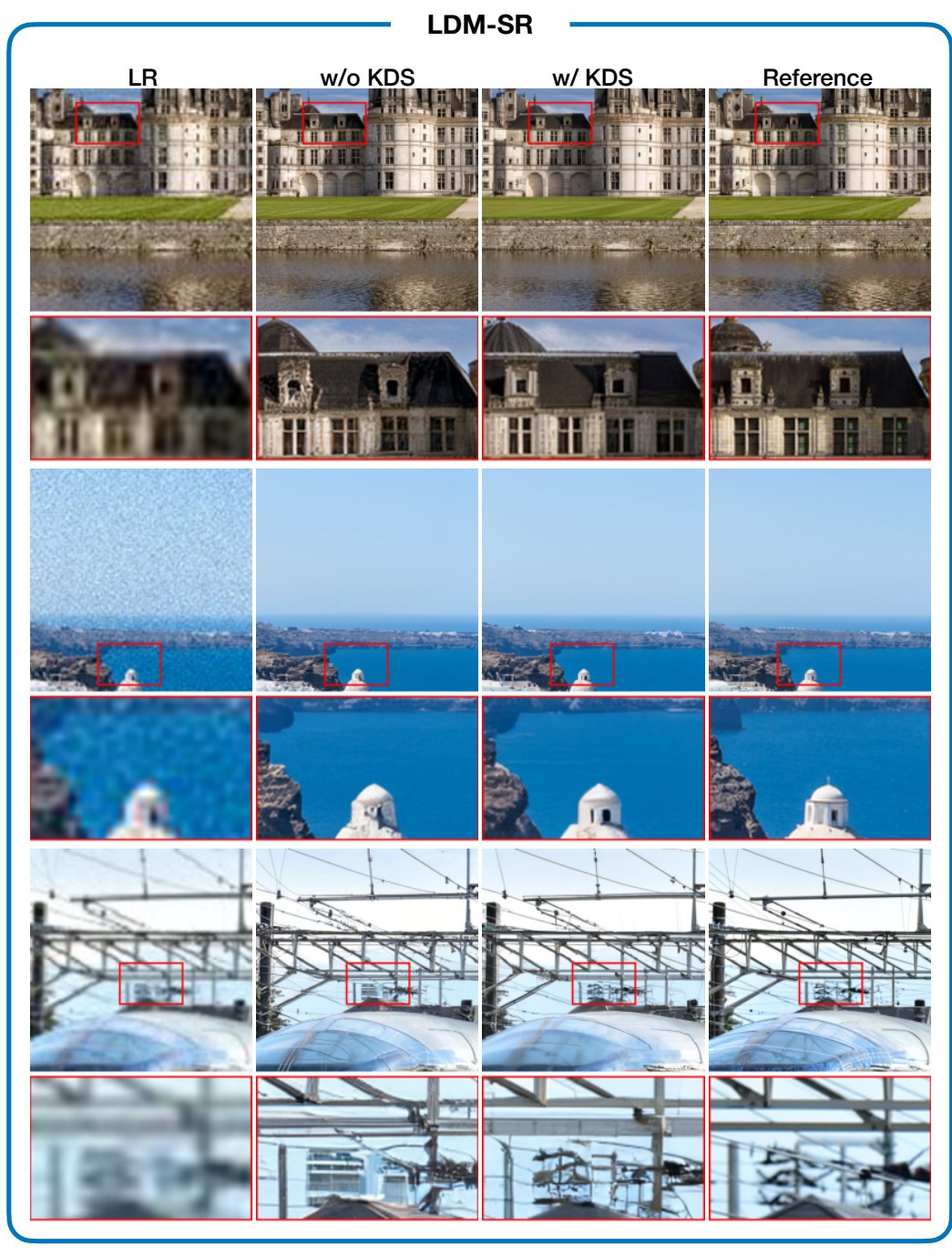

Figure 9: Real-world image super-resolution performance with LDM-SR on DIV2K dataset.

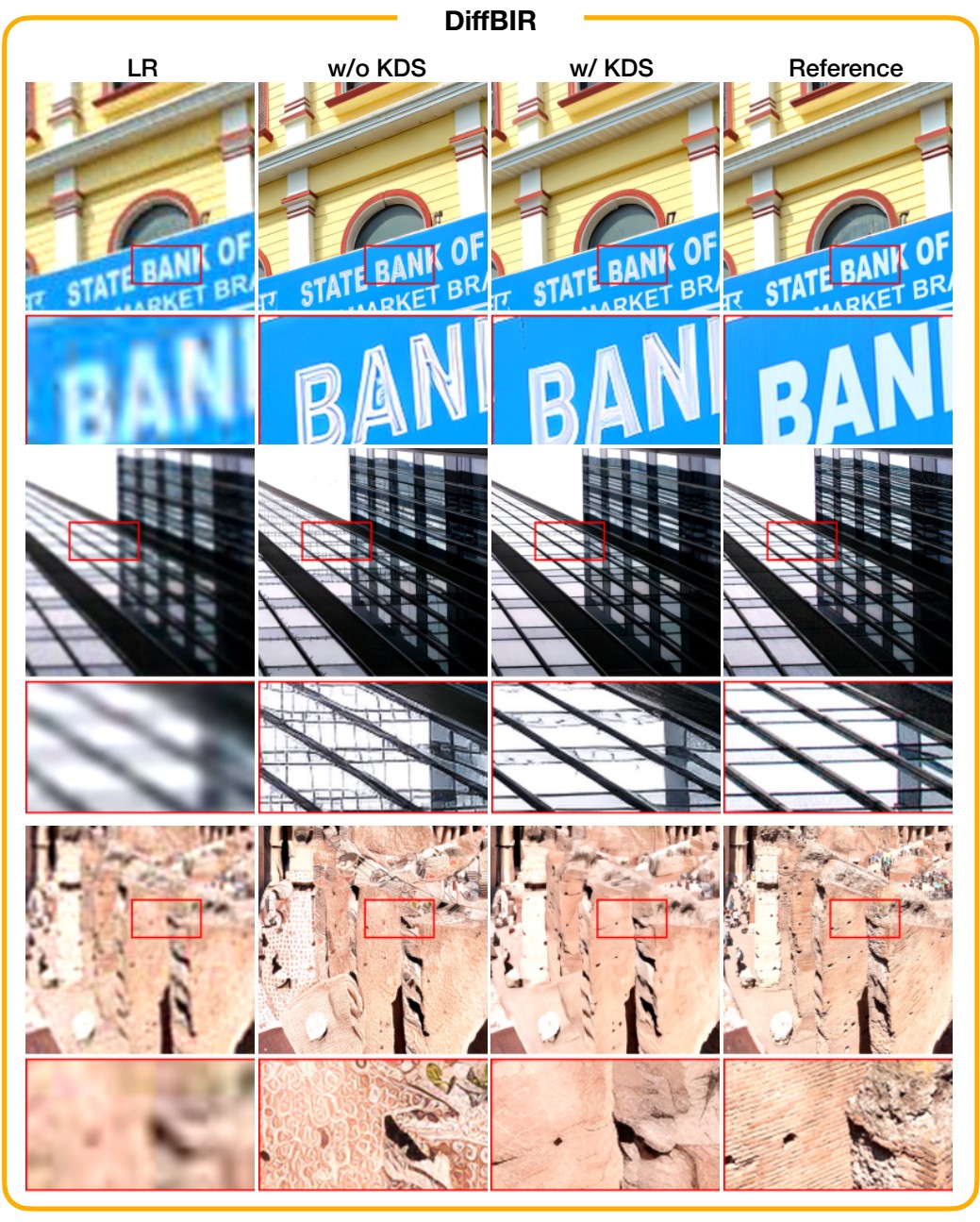

Figure 10: Real-world image super-resolution performance with DiffBIR on DIV2K dataset.

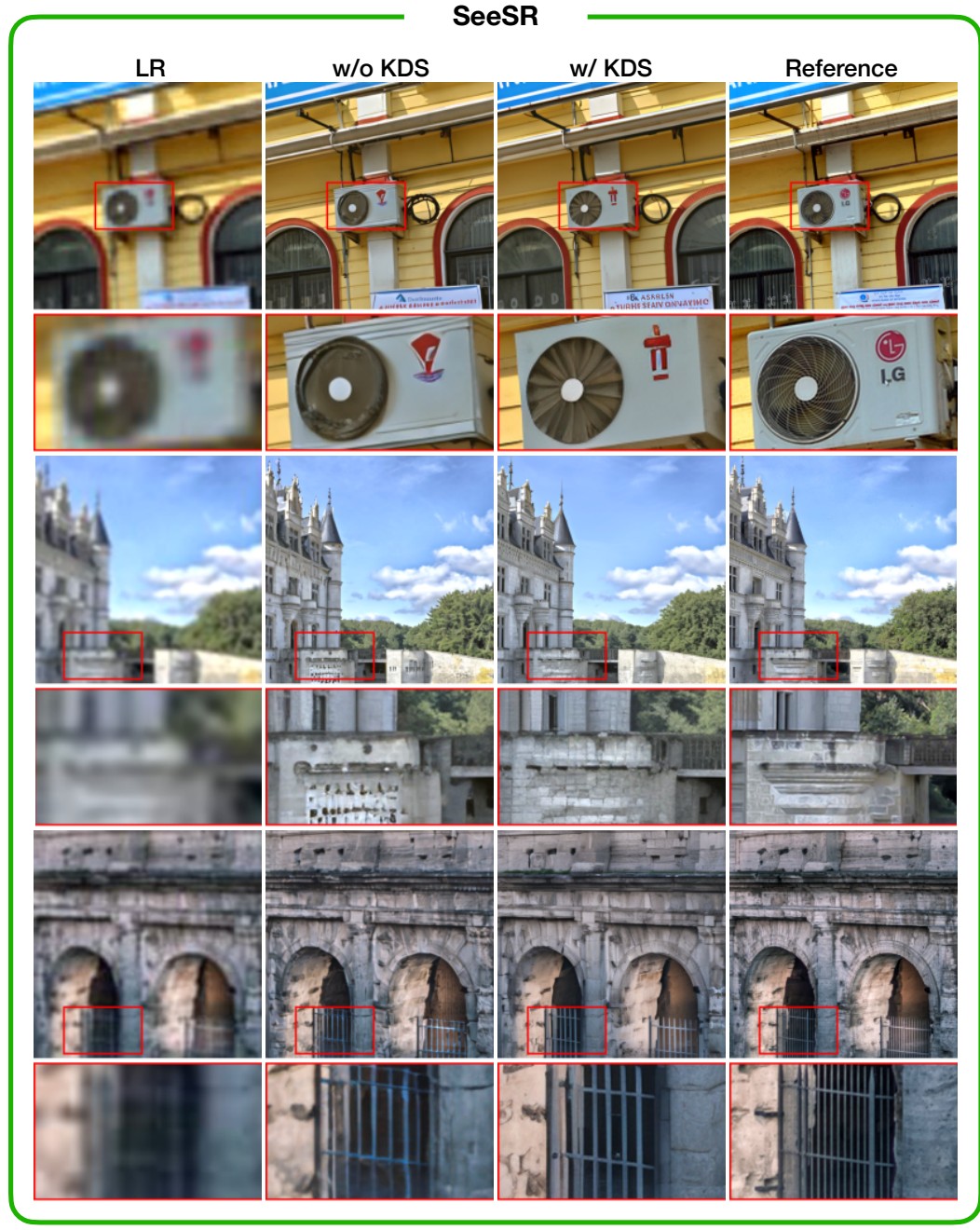

Figure 11: Real-world image super-resolution performance with SeeSR on DIV2K dataset.

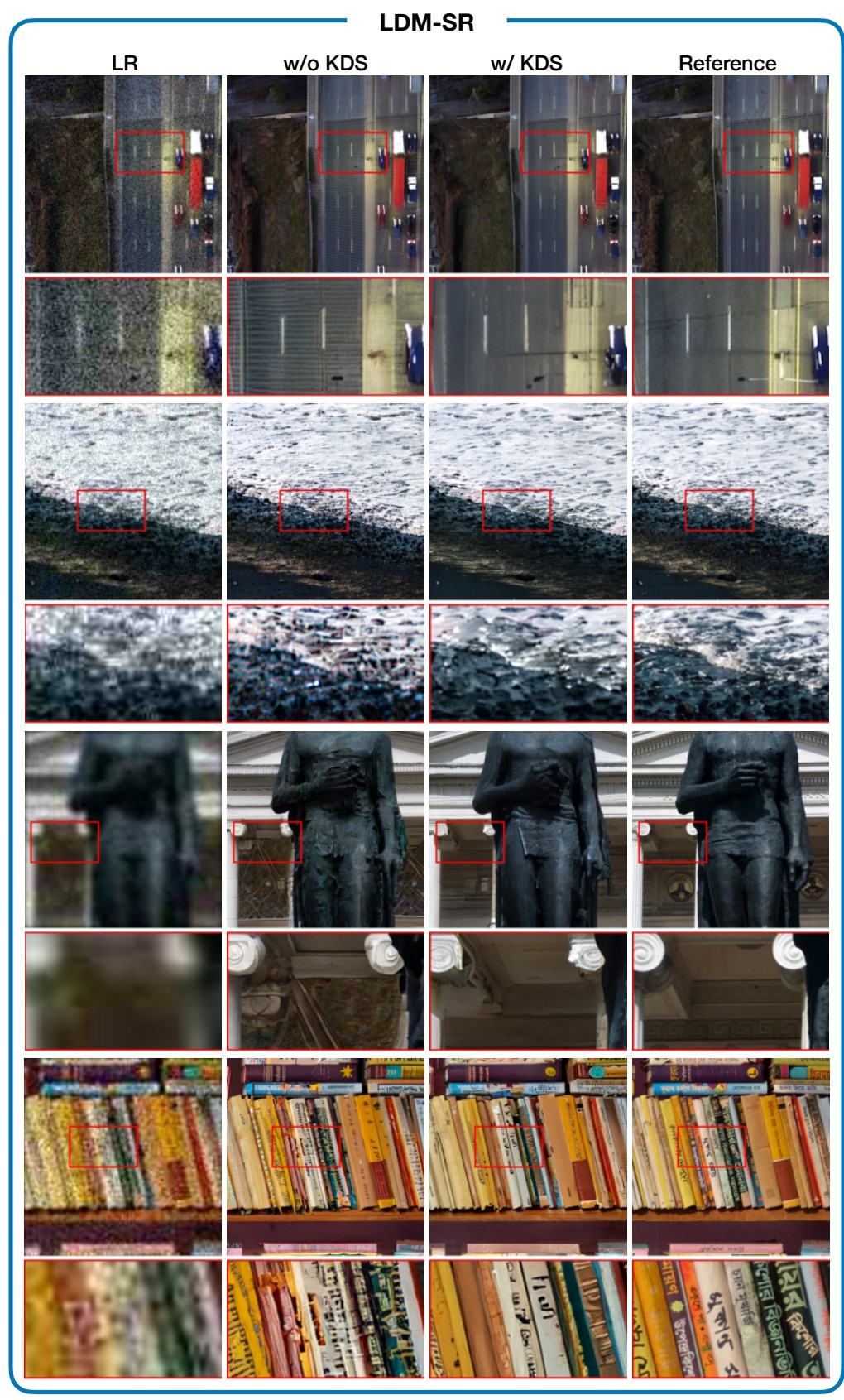

Figure 12: Real-world image super-resolution performance with LDM-SR on DF2K dataset.

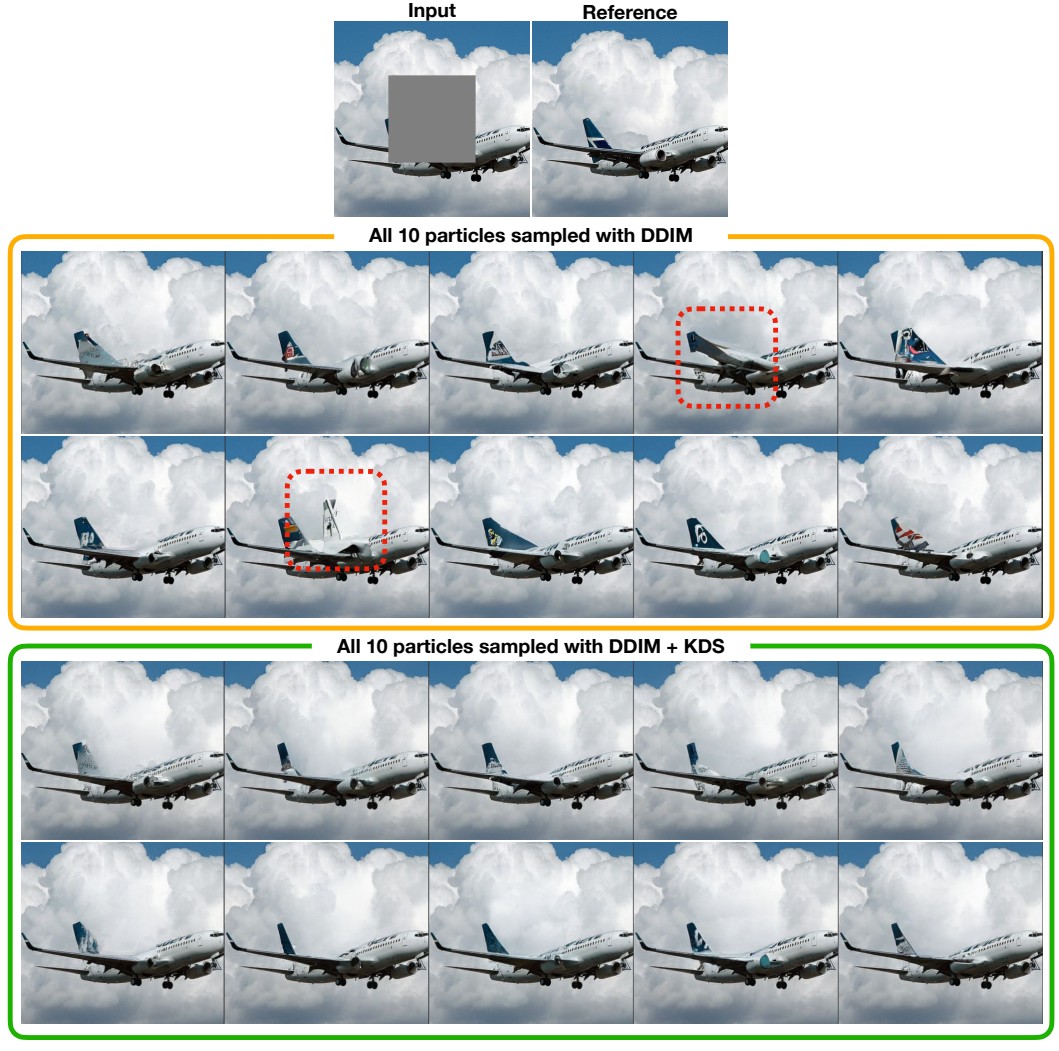

Figure 13: Image Inpainting performance with LDM-inpainting on ImageNet dataset. Visualizes all 10 particles for DDIM vs. DDIM + KDS. Regions with artifacts were highlighted with red box.

**Input**      **Reference**

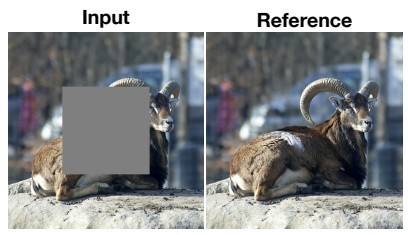

**All 10 particles sampled with DDIM**

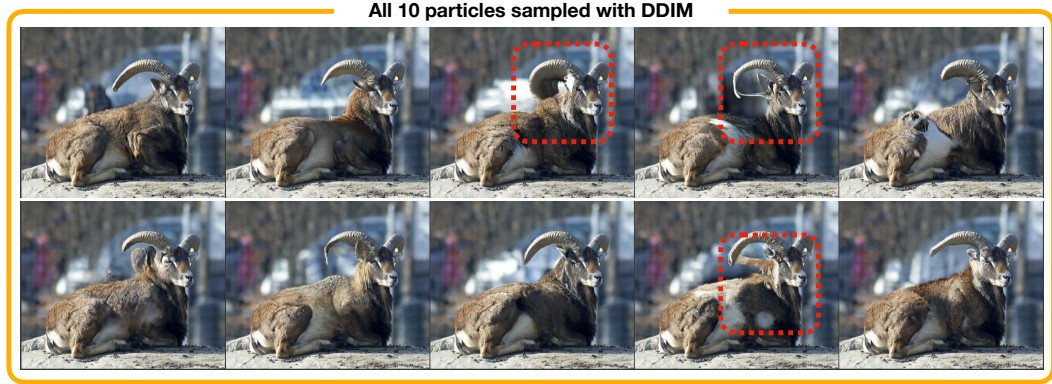

**All 10 particles sampled with DDIM + KDS**

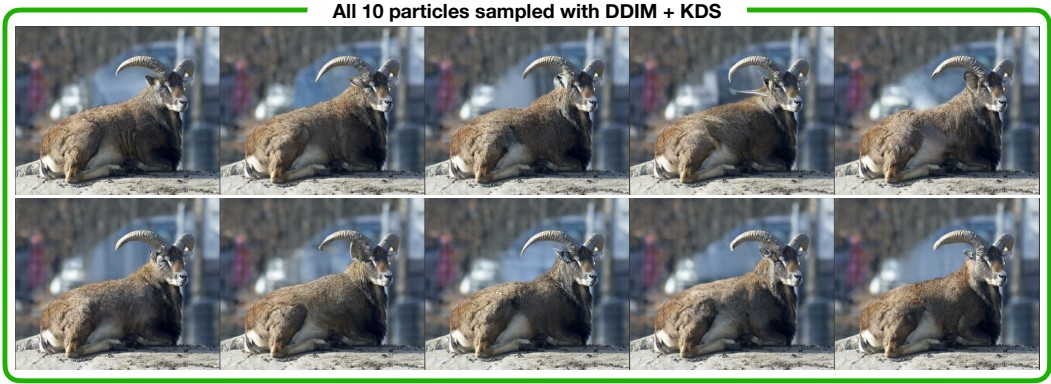

Figure 14: Image Inpainting performance with LDM-inpainting on ImageNet dataset. Visualizes all 10 particles for DDIM vs. DDIM + KDS. Regions with artifacts were highlighted with red box.

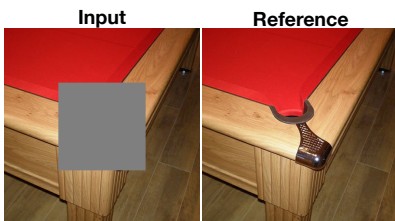

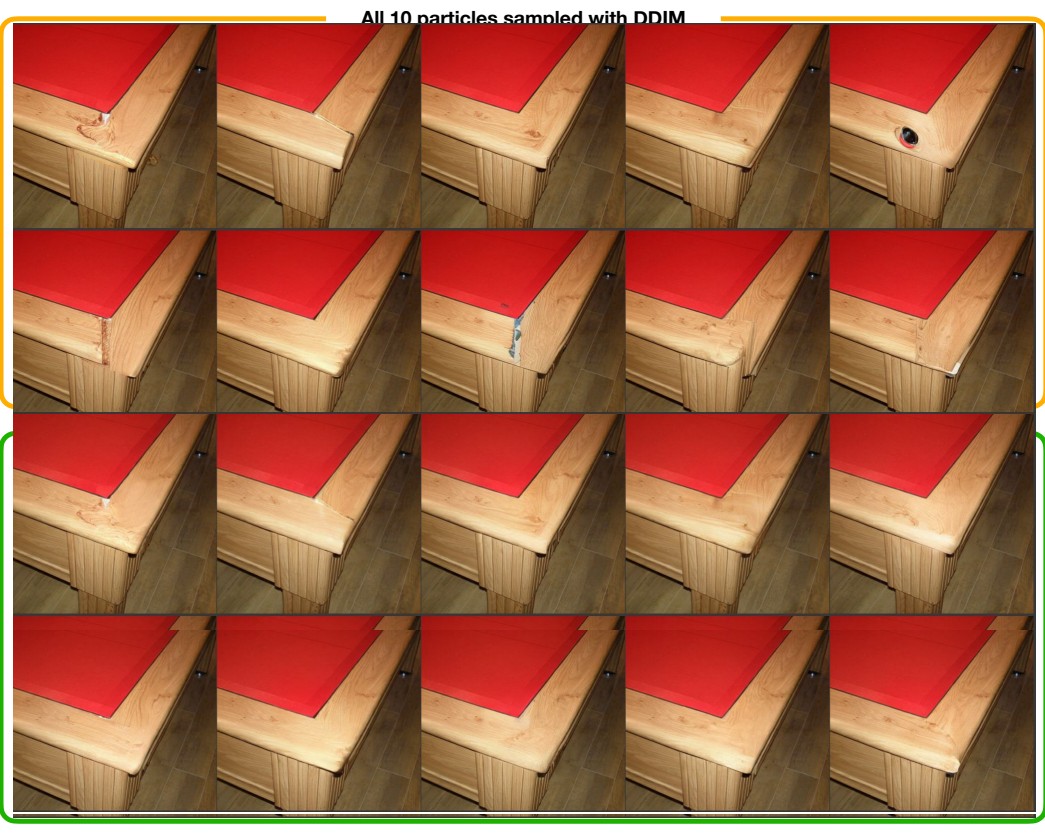

Figure 15: Image Inpainting performance with LDM-inpainting on ImageNet dataset. Visualizes all 10 particles for DDIM vs. DDIM + KDS. Regions with artifacts were highlighted with white box.

