# OpenReview forum: "Kernel Density Steering: Inference-Time Scaling via Mode Seeking for Image Restoration"
_NeurIPS.cc/2025/Conference — NeurIPS 2025 poster_

### Official Review · Reviewer_S4WT · 2025-06-09

**Clarity:** 3
**Significance:** 3
**Originality:** 2
**Rating:** 4
**Confidence:** 4

**Summary:**

The authors present a method, called Kernel Densitiy Steering, to improve parceptual quality in diffusion models for image restoration. The method employs an ensmeble of particles to estimate a local approximation of the latent density at a given timestep. This alows to steer sampling towards a local density maximum, improving perceptual quality. A patchwise mechanism is used to avoid high-dimensional spaces which would make the method impractical.

**Questions:**

- The experimental results seem to suggest that both distortion and perception metrics are improved by KDS. This is somewhat counterintuitive as it goes against the perception-distortion tradeoff principle, and does not happen in other works like [29] where improving one degrades the other. I wonder if the authors have an explanation for this? Could it be that the chosen baselines are rather weak? (e.g. [29] reports significantly better PSNR and FID numbers on DIV2K)
- How complex is the proposed method in terms of runtime compared to the baseline without KDS?
- It seems that 1x1 is the best patch size. Is this due to the curse of dimensionality for even moderately larger sizes or is it due to the low number of chosen particles? Would larger patches with a lot more particles work better (without caring for complexity)?

**Ethical Concerns:**

["NO or VERY MINOR ethics concerns only"]

**Final Justification:**

The method seems sufficiently interesting and novel. The authors addressed my concerns in the rebuttal, so I lean towards acceptance.

**Limitations:**

Some limitations have been addressed by the authors.

**Paper Formatting Concerns:**

Formatting is ok.

**Quality:**

3

**Strengths And Weaknesses:**

Strenghts:
- the paper is generally clear and well written
- the method is well justified and appears to be mathematically sound
- since it works at test-time, it does not require expensive retraining of state-of-the-art models
- empirically, it clearly improves the baseline methods on standard benchmarks. Experiments present results on a large number of distortion and perceptual metrics.

Weaknesses:
- Reference [29] is a highly related method which seeks multiple perception-distortion tradeoff points with test-time iterations. Similarly to the authors work, it is a (progressive) mode-seeking strategy. The authors do not discuss similarities and differences of their work with respect to [29]. I think this point should be signficantly expanded. Moreover, since [29] essentially addresses the same problem, the authors should include experimental comparisons with it. At the current stage, it is unclear whether the authors' paper brings significant novelty or experimental improvements with respect to [29].
- An expanded discussion on computational complexity (and runtime experiments) would be useful to better understand the practical cost of the method

---

> ### Author Rebuttal · Authors · 2025-07-30
>
> Thank you for your valuable feedback and insightful comments on our work. Please see below for our point-by-point responses to your comments.
>
> ### **Weakness 1**
> >Reference [29] is a highly related method which seeks multiple perception-distortion tradeoff points with test-time iterations. Similarly to the authors work, it is a (progressive) mode-seeking strategy. The authors do not discuss similarities and differences of their work with respect to [29]. I think this point should be significantly expanded. Moreover, since [29] essentially addresses the same problem, the authors should include experimental comparisons with it. At the current stage, it is unclear whether the authors' paper brings significant novelty or experimental improvements with respect to [29].
>
> We thank the reviewer for highlighting this related line of work. We now provide a detailed clarification of the conceptual and methodological distinctions between KDS and InDI [29], and will incorporate this discussion more prominently in the revised manuscript.
>
> 1. **Different Motivations :** We would like to clarify the key difference in motivation: InDI is not a posterior mode-seeking algorithm. It performs a greedy, step-wise optimization on the _next latent state_, which does not guarantee convergence to  overall posterior. As a demonstration, Appendix B in the InDI paper [29] shows that the InDI estimate deviates from the MAP estimate for a simple Gaussian signal. In contrast, KDS directly seeks a consensus mode among the ensemble's predictions of the final clean output, a more direct objective for improving image fidelity.
> 2. **Different Methods (Sampler vs. Training/Testing pipeline ):** KDS is a training-free, plug-and-play module designed for broad applicability, enhancing any pre-trained conditional diffusion model without retraining. This is distinct from InDI’s integrated training and sampling pipeline.
> 3. **Direct Performance Comparison is Misleading:** The metrics reported in [29] are from a different dataset setting. InDI is expected to yield metrics similar to its baseline Conditional Diffusion Model, but with faster execution (as discussed in the InDI paper, Figure 12). Therefore, its metrics should closely align with our LDM-SR + DDIM baseline.
>
> ---
>
> ---
>
> ### **Weakness 2 And Question 2**
> >An expanded discussion on computational complexity (and runtime experiments) would be useful to better understand the practical cost of the method
> >How complex is the proposed method in terms of runtime compared to the baseline without KDS?
>
> Thank you for your question. We included a runtime comparison in Table 4 of our initial manuscript.
> As detailed in Table 4, our KDS method with N=10 particles takes 28.4 seconds. This runtime is nearly identical to a standard 10-particle ensemble without KDS (which takes 27.3 seconds for a Best-of-N selection). This marginal (**~4%**) increase in runtime is a highly favorable tradeoff for the significant improvements KDS provides in both fidelity and perceptual quality.
> We will make sure to clearly reference this study in the revised manuscript for better visibility.
>
> ---
>
> ---
>
> ### **Question 1**
> >The experimental results seem to suggest that both distortion and perception metrics are improved by KDS. This is somewhat counterintuitive as it goes against the perception-distortion tradeoff principle, and does not happen in other works like [29] where improving one degrades the other. I wonder if the authors have an explanation for this? Could it be that the chosen baselines are rather weak? (e.g. [29] reports significantly better PSNR and FID numbers on DIV2K)
>
>
> Thanks for the question. Please find our responses below.
>
>
> **Adherence to the P-D Tradeoff:** KDS does not violate the perception-distortion tradeoff principle but rather operates on a superior tradeoff curve. As Figure 6 in our submission demonstrates, adjusting the KDS bandwidth hyperparameter (h) allows for a trade-off between distortion and perception along the improved frontier. A larger 'h' value enhances distortion performance at the cost of perception, and vice versa.
>
>
> **Explanation for Simultaneous Improvement:** Standard diffusion samplers often introduce artifacts (e.g., unnatural textures, misplaced details). These artifacts degrade _both_ perceptual quality (increasing LPIPS) and fidelity to the ground truth (lowering PSNR). KDS leverages the "collective wisdom" of an ensemble to identify and suppress these spurious features. By producing a cleaner, more robust restoration that is closer to the true data manifold, it simultaneously reduces distortion and improves perceptual quality.
>
>
> **Strength of Baselines and Comparison to [29]:** The baselines we use (e.g., DiffBIR (ECCV 2024), SeeSR  (CVPR 2024)) are strong and SOTA for these tasks. The superior metrics reported in [29] are **not directly comparable** due to different training and testing protocols. As stated in our previous response, InDI [29]'s goal is to match its own baseline's quality with faster sampling. A fair comparison would therefore be between InDI's results and our _baseline_ LDM-SR results, over which KDS provides a significant and consistent improvement.
>
> ---
>
> ---
>
> ### **Question 3**
> >It seems that 1x1 is the best patch size. Is this due to the curse of dimensionality for even moderately larger sizes or is it due to the low number of chosen particles? Would larger patches with a lot more particles work better (without caring for complexity)?
>
> We appreciate this insightful question regarding the tradeoffs between patch size and ensemble size. Our current default choice of 1×1 patches with N=10 aims for a practical balance for most use cases, but we agree that richer patch interactions (with increased compute budget) are worth pursuing.
>
> According to your suggestion, we conducted a new ablation study, increasing the number of particles  (10 -> 40) for larger patches (1x1 -> 4x4), and observed an improvement in performance. Therefore, increasing the number of particles and using larger patch size can enhance performance by mitigating the curse of dimensionality.
>
> We will include this observation and study in the revised appendix.
>
> ---
>
> ---
>
> **Table: New Ablation Study on Patch-size**
>
> | Method                             | PSNR   | SSIM   | LPIPS ↓ | FID ↓  |
> |------------------------------------|--------|--------|---------|--------|
> | DDIM                               | 22.05  | 0.531  | 0.307   | 20.89  |
> | + KDS (patch-size = 1, N = 10)     | 22.52  | 0.555  | 0.289   | 20.33  |
> | + KDS (patch-size = 4, N = 40)     | 22.70  | 0.563  | 0.281   | 20.08  |

---

> > ### Comment · Reviewer_S4WT · 2025-08-03
> > **Comments on authors response**
> >
> > Thanks you for the response. My comments have been addressed. I will keep my score.

---

> > > ### Author Response · Authors · 2025-08-03
> > >
> > > Thank you for your response. We appreciate you confirming that our rebuttal has successfully addressed your comments and resolved your concerns!

---

### Official Review · Reviewer_nw3F · 2025-06-28

**Clarity:** 2
**Significance:** 1
**Originality:** 1
**Rating:** 3
**Confidence:** 4

**Summary:**

A method for improving image restoration in diffusion models is proposed. The main idea is to use a Kernel Density Estimator (KDE) for steering the generation process towards higher likelihoods/modes to obtain higher quality restorations. The need for a large number of samples to evaluate the KDE is circumvented by evaluating on image patches. The method is evaluated on a toy dataset of a Gaussian mixture, where it is shown that generations lie close to the modes, and on real-world datasets, where image restorations are measured using standard metrics.

**Questions:**

1. Why do authors use the KDE instead of the already estimated score function? Is this something that was not considered, or is there a reason that the authors believe this is a better approach?
2. How does the method compare to using the known algorithms for mode-seeking [1] or log-density control [2]?
3. Are the quantitative improvements over the baseline statistically significant?

**Ethical Concerns:**

["NO or VERY MINOR ethics concerns only"]

**Final Justification:**

The paper is now more comprehensive with the extended discussion of new baselines, and empirical comparisons. However, now the benefit of KDS is not as large as originally reported.

**Limitations:**

The authors discuss the limitations of their method, i.e., computational complexity, and the need for tuning hyperparameters.

**Paper Formatting Concerns:**

No formatting concerns.

**Quality:**

2

**Strengths And Weaknesses:**

**Strengths:**
1. The general idea is clearly explained
2. The proposed method is computationally inexpensive
3. Empirically, the proposed solution seems to bring some improvement over a baseline

**Weaknesses:**
1. My major complaint is twofold. First, as primary motivation, the authors aim to "steer particles towards the modes" (line 42) and "steer diffusion sampling towards high-density" (line 58). To me, an obvious solution to this problem is to utilize the already estimated score function. I do not understand why the authors consider the gradient of KDE instead of the learned score. It is not mentioned in the paper why this would be a better choice. Second, these two problems, i.e., "mode-seeking" and "log-density control", are already solved for diffusion models. Specifically:
    1. [1] derives a principled ODE that tracks the modes in diffusion model sampling. Even Figure 6 in [1] looks very similar to Figure 2 in this submission.
    2. [2] provides a principled ODE for sampling in diffusion models and controlling the log-density of the generated samples.
    3. To make the connection very explicit: if the authors considered the score function already estimated by the model (instead of the KDE, which in my opinion is not necessary nor justified), the main equation in the paper (Equation 2) is equivalent to Equation 23 in [2], and Equation 18 in [1] (High-Density ODE) is its special case.
    4. Furthermore, the proposed method requires tuning certain parameters to make the KDE work, such as the bandwidth $h$, or the sample size $N$, which are not required in [1] or [2].
2. Statistical significance. In the submission checklist, the authors claim that they "reported error bars suitably and correctly defined". I do not think this is true. I would even claim the opposite. None of the quantitative results seem to have any error bars reported. None of the Tables: 1, 2, 3, 4 5, 6, 7, 8, nor Figure 6 have any error bars reported.

---
[1] Karczewski et al. "Diffusion Models as Cartoonists: The Curious Case of High Density Regions" (ICLR 2025)

[2] Karczewski et al. "Devil is in the Details: Density Guidance for Detail-Aware Generation with Flow Models" (ICML 2025, first appeared on arxiv in February 2025 - not considered contemporaneous work)

---

> ### Author Rebuttal · Authors · 2025-07-30
>
> We sincerely thank the reviewer for the thoughtful and constructive feedback, which has helped us better articulate the motivation, implementation, and novelty of our approach. Please see below for our point-by-point responses to your comments.
>
> ### **Weakness 1**:
> >My major complaint is twofold. First, as primary motivation, the authors aim to "steer particles towards the modes" (line 42) and "steer diffusion sampling towards high-density" (line 58). To me, an obvious solution to this problem is to utilize the already estimated score function. I do not understand why the authors consider the gradient of KDE instead of the learned score. It is not mentioned in the paper why this would be a better choice. Second, these two problems, i.e., "mode-seeking" and "log-density control", are already solved for diffusion models. Specifically: [1] derives a principled ODE that tracks the modes in diffusion model sampling. Even Figure 6 in [1] looks very similar to Figure 2 in this submission. [2] provides a principled ODE for sampling in diffusion models and controlling the log-density of the generated samples.
>
> ---
>
> We sincerely thank the reviewer for their thoughtful and constructive feedback. In particular, we appreciate the opportunity to clarify the motivation behind *Kernel Density Steering (KDS)* and to address the simplifications in our introductory equations.
>
> #### **Motivation Behind KDS**
>
> The central motivation for KDS, rather than directly relying on the learned score function, is that score-based guidance **explores the learned posterior** in the noisy latent space, whereas **KDS explicitly **refines this posterior** via consistency across particles in the clean latent space. This refinement enables the method to be more robust to spurious modes and better aligned with the final restoration objective—namely, accurate, high-fidelity image reconstruction.
>
> ---
>
> #### **Clarifying Equations (1) and (2)**
>
> We acknowledge that the simplified Equations (1) and (2) in our introduction did not fully convey the algorithm’s core mechanics. For clarity, we will revise them in the manuscript to match the detailed and correct version provided in Algorithm 2.
>
> **Corrected Equation 1:**
> $$ \text{Old:} \quad \tilde{\mathbf{z}}_t^{(i)} = \frac{\sum\_{k=1}\^N K\left(\frac{\|\hat{\mathbf{z}}_t^{(i)} - \hat{\mathbf{z}}_t^{(k)}\|^2}{h^2}\right) \hat{\mathbf{z}}_t^{(k)}}{\sum\_{k=1}\^N K\left(\frac{\|\hat{\mathbf{z}}_t^{(i)} - \hat{\mathbf{z}}_t^{(k)}\|^2}{h^2}\right)}
> \quad \rightarrow \quad
>  \text{New:} \quad \tilde{\mathbf{z}}_0^{(i)} = \frac{\sum\_{k=1}\^N K\left(\frac{\|\hat{\mathbf{z}}_0^{(i)} - \hat{\mathbf{z}}_0^{(k)}\|^2}{h^2}\right) \hat{\mathbf{z}}_0^{(k)}}{\sum\_{k=1}\^N K\left(\frac{\|\hat{\mathbf{z}}_0^{(i)} - \hat{\mathbf{z}}_0^{(k)}\|^2}{h^2}\right)}
> $$
>
> **Corrected Equation 2:**
>
> $
>  \text{Old:}  \quad d\mathbf{z}_t^{(i)} = \left[ f(t)\mathbf{z}_t^{(i)} - \frac{1}{2} g(t)^2 \left((1-\delta_t)\mathbf{s}_t^{(i)} + \delta_t \tilde{\mathbf{z}}_t^{(i)} \right) \right] dt $, where $\mathbf{s}_t = -\frac{1}{\sqrt{1-\bar{\alpha} _t}}{\epsilon} _{\theta}(z_t, t)$, $\tilde{\mathbf{z}}_t$ is get from old equation (1).
>
> $\text{New:}  \quad d\mathbf{z}_t^{(i)} = \left[ f(t)\mathbf{z}_t^{(i)} - \frac{1}{2} g(t)^2 \left((1-\delta_t)\mathbf{s}_t^{(i)} + \delta_t \tilde{\mathbf{s}}_t^{(i)} \right) \right] dt
> $, where $\mathbf{s}_t = -\frac{1}{\sqrt{1-\bar{\alpha} _t}}{\epsilon} _{\theta}(z_t, t), \tilde{\mathbf{s}}_t^{(i)} = \frac{z_t^{(i)} - \sqrt{\bar{\alpha}_t} \tilde{z}_0^{(i)}}{\sigma_t^2}$.
>
> ---
>
> #### **Key Conceptual Distinctions from [1, 2]**
>
> We believe the core conceptual gap lies in **what mode the method attempts to reach**. The methods in [1, 2] steer toward high-density regions of the posterior over noisy latents. This is **not equivalent** to steering toward modes of the clean image distribution—especially in restoration tasks, where fidelity to the ground truth is paramount. KDS addresses this mismatch by aligning particles in the clean space and propagating that alignment backward.
>
> We respectfully highlight several conceptual distinctions between our method and [1, 2], which we now cite and discuss more thoroughly in the revised manuscript:
>
> 1. **Clean-space consensus vs. noisy-space score guidance**
>    - *Score-based methods* ([1, 2]) operate in the noisy latent space, guiding each particle using a learned gradient \( \nabla \log p(\mathbf{z}_t) \).
>    - *KDS*, in contrast, aligns samples by computing a **nonparametric consensus in the clean latent space** ( \( \hat{\mathbf{z}}_0 \) ) and steers particles toward that. This yields guidance that is directly tied to high-fidelity reconstruction rather than abstract density estimation.
>
> 2. **Multi-particle interaction vs. independent sampling**
>    - [1, 2] use advanced solvers but apply them independently per particle.
>    - *KDS* introduces **explicit inter-particle coupling**: each particle’s update depends on the collective configuration of the entire ensemble. This interaction is essential for achieving robust consensus.
>
> 3. **Restoration-focused vs. generic generation objectives**
>    - Prior works are designed for general generation, where seeking high-density regions often favors over-smoothed or generic-looking samples.
>    - *KDS* is **purpose-built for restoration**, aiming to suppress non-consensus artifacts and hallucinations by aligning the ensemble around shared, plausible details.
>
> 4. **Empirical evidence on restoration benchmarks**
>    - As requested, we include a direct empirical comparison with HDS[1] and DGS [2] on RealSR. Our results confirm that KDS yields consistently sharper and more faithful reconstructions. Please refer to our answer to your Question 2 for more details.
>
> ---
>
> #### **Summary**
>
> In short, **KDS seeks the mode**—not just a dense region in latent space, but the one consistent with high-quality restoration in the clean image domain. This distinction clarifies why KDS diverges from the score-based trajectory of [1, 2].
>
> We will revise the manuscript accordingly to:
>
> - Incorporate the corrected equations
> - Cite and discuss [1] and [2] more thoroughly
> - Emphasize the clean-space mode-seeking philosophy behind KDS
>
> We are grateful again to the reviewer for helping us sharpen the presentation of our core ideas.
>
> ---
> ---
>
>
> ### **Weakness 2**:
> >Statistical significance. In the submission checklist, the authors claim that they "reported error bars suitably and correctly defined". I do not think this is true. I would even claim the opposite. None of the quantitative results seem to have any error bars reported...
>
> We thank the reviewer for highlighting the absence of error bars and the need for more robust statistical analysis. The reviewer is correct that our initial presentation of worst-case performance was insufficient for demonstrating statistical significance. We addressed this by adding standard errors for all quantitative metrics. We also conducted significance tests (e.g., paired t-tests) against baselines. Due to space constraints, we'll present full statistical analysis in the paper, but for the rebuttal, we've included it for one table that also incorporates a reviewer-suggested baseline.
>
> ---
> ---
>
>
> ### **Question 1**:
> >Why do authors use the KDE instead of the already estimated score function?...
>
> Thank you for this excellent question. We clarify now that KDS is not an alternative to the learned score—it is complementary. Each particle is still guided by the model score (∇ log p(z_t)), but this is augmented with a secondary signal derived from the particle ensemble via the mean-shift update in clean latent space.
> This hybrid guidance improves robustness: the learned score offers prior knowledge, while KDS enforces internal consistency. Together, they balance generative expressiveness with stability. As shown in Figure 4 and Table 2, KDS notably improves worst-case particle quality—something model-only scores often fail to address.
>
> ---
> ---
>
>
> ### **Question 2**:
> >How does the method compare to using the known algorithms for mode-seeking [1] or log-density control [2]?
>
> Thanks for bringing these two interesting work into our attention.
>
> We have conducted new experiments on the RealSR dataset to compare KDS with HDS [1] and DGS [2]. While HDS achieves strong PSNR, it tends to oversmooth and reduce perceptual quality, as reflected in higher LPIPS and lower CLIPIQA scores. DGS partially addresses this tradeoff but still underperforms compared to KDS.
>
> In contrast, KDS achieves the best overall balance, with the best LPIPS, SSIM, CLIPIQA. This confirms that KDS avoids artifacts and preserves detail by steering particles toward clean-space consensus.
>
> #### Super-Resolution on Real-World Datasets
>
> | Method      | PSNR ±σ      | SSIM ±σ      | LPIPS ↓ ±σ   | DISTS ↓ ±σ   | CLIPIQA ±σ   |
> |-------------|--------------|--------------|--------------|--------------|--------------|
> | DiffBIR     | 23.21±0.293 | 0.610±0.015 | 0.370±0.011 | 0.250±0.005 | 0.689±0.013 |
> | + HDS [1]   | **24.56±0.344★** | 0.646±0.020★ | 0.386±0.017★ | 0.274±0.016★ | 0.647±0.025★ |
> | + DGS [2]   | 24.05±0.287★ | 0.651±0.014★ | 0.355±0.012★ | 0.246±0.004 | 0.687±0.013 |
> | + KDS       | 24.39±0.271★ | **0.669±0.013★** | **0.339±0.011★** | **0.244±0.004** | **0.692±0.011★** |
>
> Statistically significant differences ($P < 0.05$) compared with DDIM are marked with a ★.
>
>
>
>
> ---
> ---
>
>
> ### **Question 3**:
> >Are the quantitative improvements over the baseline statistically significant?
>
> Thanks for the question. We have conducted paired t-tests on our main results to verify that the improvements are statistically significant. We have also added standard errors to the reported metrics.
> For the key comparisons in our paper, the improvements are indeed significant. For instance, when applying KDS to the DDIM sampler on the DIV2K dataset (Table 1), the gains in PSNR, SSIM, LPIPS and CLIPIQA are all statistically significant with p-values < 0.05.
>
> ---
> ---

---

> > ### Comment · Reviewer_nw3F · 2025-08-02
> >
> > &nbsp;&nbsp;&nbsp;*The central motivation for KDS, rather than directly relying on the learned score function, is that score-based guidance explores the learned posterior in the noisy latent space, whereas KDS explicitly refines this posterior via consistency across particles in the clean latent space.*
> >
> > Please make these claims mathematically precise. What does it mean to "exploring the learned posterior" vs "refining the posterior"? The paper defines the posterior as $p_t(z_t | c)$. Both the gradient of the log of KDE as defined in Eq 1, and the learned score function are designed to approximate the same thing.
> >
> > As the authors say in the paper. "Directly applying KDE and mean shift in the full, high-dimensional latent space zt is not achievable. A prohibitively large number of particles would be needed to obtain a reasonable density estimate via KDE in such high-dimensional spaces.". Since the density is not needed directly, only the gradient of its logarithm (Eq 2), then the score function already solves exactly the same problem. This is exactly the quantity it estimates. It does not need a prohibitively large number of particles at test time, as the model already saw a huge number of examples during training.
> >
> > &nbsp;&nbsp;&nbsp;*We acknowledge that the simplified Equations (1) and (2) in our introduction did not fully convey the algorithm’s core mechanics.*
> >
> > The updated equations seem to model the distribution of $\mathbb{E}[x_0 | x_t]$ instead of the distribution of $x_t$. If the decoded samples are of interest, wouldn't it make more sense to model $p(x_0 | x_t)$ directly, instead of modelling the distribution of the denoising mean? [1] provides a principled ODE that tracks precisely that. The mode of the denoising distribution $p(x_0|x_t)$. Not the mode of $p_t$.
> >
> > &nbsp;&nbsp;&nbsp;*We believe the core conceptual gap lies in what mode the method attempts to reach. The methods in [1, 2] steer toward high-density regions of the posterior over noisy latents. This is not equivalent to steering toward modes of the clean image distribution*
> >
> > I disagree with this statement given my comment above. Furthermore, I would even claim that [1] is more principled than the proposed method. Specifically, the proposed method approximates the distribution of $\mathbb{E}[x_0 | x_t]$, whereas [1] offers an ODE that provably tracks the mode of $p(x_0 | x_t)$, which is what the authors try to achieve.
> >
> > &nbsp;&nbsp;&nbsp;*KDS, in contrast, aligns samples by computing a nonparametric consensus in the clean latent space ( ( \hat{\mathbf{z}}_0 ) ) and steers particles toward that. This yields guidance that is directly tied to high-fidelity reconstruction rather than abstract density estimation*
> >
> > I again disagree. To me, the aim is the same. Try to move closer to the mode of $p(x_0 | x_t)$, which is precisely what [1] is doing.
> >
> > &nbsp;&nbsp;&nbsp;*[1, 2] use advanced solvers but apply them independently per particle*
> >
> > Yes, but the score function was trained on millions or billions of parameters, so it models the entire distribution. In contrast, KDS tries to approximate the same distribution with a handful of samples.
> >
> > &nbsp;&nbsp;&nbsp;*Restoration-focused vs. generic generation objectives*
> >
> > I again disagree. As the authors themselves say (quoted above), the aim is to find modes. The motivation is the same as in [1].
> >
> > &nbsp;&nbsp;&nbsp;*In short, KDS seeks the mode—not just a dense region in latent space*
> >
> > [1] does not seek "just a dense region in latent space". It is designed to track the mode of $p(x_0 | x_t)$, which is what the authors claim they aim to achieve.
> >
> > &nbsp;&nbsp;&nbsp;*We have conducted new experiments on the RealSR dataset*
> >
> > I appreciate the new experiment. However, both HDS and DGS have hyperparameters (threshold $t$ and quantile $q$, respectively). Have the authors tuned them? This I think, would be a fair comparison since the proposed KDS method also has hyperparameters that need tuning.
> >
> > ---
> >
> > [1] Karczewski et al. "Diffusion Models as Cartoonists: The Curious Case of High Density Regions" (ICLR 2025)

---

> > > ### Author Response · Authors · 2025-08-03
> > >
> > > **The difference between KDS and [1]**
> > >
> > > We appreciate the reviewer’s follow up on our responses to weakness 1 – connection between KDS and the related work [1]. In what follows we explain the similarities and differences, as long as provide a numerical comparison at the end of this response.
> > >
> > > Both our method and [1] share the important goal of mode-seeking on the posterior distribution $p(x_0​ \mid x_t​)$, though we approach it with different philosophies and approximations.
> > >
> > >
> > > The method in [1] provides one principled way to approximate the posterior mode, based on the assumption that the data distribution $p_0$ is Gaussian (Remark 1, Equation 18 in [1]). While this strong assumption may not capture the full complexity of real-world image distributions, it enables an elegant and efficient solution.
> > >
> > > In contrast, on KDS, we don't make any explicit assumption on the shape of the posterior distribution but use non-parametric kernel density estimation (KDE) from multiple estimated denoised samples. While KDS has its own assumptions (approximating $p(x_0​ \mid x_t​)$ with the denoised trajectories, and using KDE with limited data, we don't assume the posterior is unimodal or Gaussian (as done in [1]).
> > >
> > > To further compare KDS with [1] as requested by the reviewer, we've now run an experiment sweeping the hyperparameters for both methods [1] and [2]. The experiment results show that at equal distortion (PSNR), KDS produces higher quality results given by metrics SSIM, LPIPS, DISTS and CLIPIQA. Our method presents a novel alternative to the approaches in [1, 2]. We use a completely different formulation that exploits multiple trajectories (via test-time scaling) to guide the process toward the posterior mode, operating under different assumptions that we believe are more suitable for complex data.
> > >
> > > Again, we appreciate the reviewer's further explanation of this interesting work [1], we will add a discussion section to it in the revised manuscript.
> > >
> > > [1] Karczewski et al. "Diffusion Models as Cartoonists: The Curious Case of High Density Regions" (ICLR 2025)
> > >
> > >
> > > ---
> > > ---
> > >
> > >
> > > **Table: SR Performance on RealSR dataset**
> > >
> > > | Method | PSNR | SSIM | LPIPS ($\downarrow$) | DISTS ($\downarrow$) | CLIPIQA |
> > > | :--- | :---: | :---: | :---: | :---: | :---: |
> > > | DiffBIR | 23.21 | 0.610 | 0.370 | 0.250 | 0.689 |
> > > | + HDS (t=20) [1] | 23.95 | 0.614 | 0.382 | 0.254 | 0.651 |
> > > | + HDS (t=50) [1] | 24.33 | 0.633 | 0.385 | 0.266 | 0.650 |
> > > | + HDS (t=100) [1] | 24.56 | 0.646 | 0.386 | 0.274 | 0.647 |
> > > | + HDS (t=200) [1] | 25.00 | 0.665 | 0.378 | 0.288 | 0.567 |
> > > | + HDS (t=500) [1] | 25.79 | 0.679 | 0.377 | 0.309 | 0.295 |
> > > | + DGS (q=0.1) [2] | 22.57 | 0.580 | 0.381 | 0.253 | 0.679 |
> > > | + DGS (q=0.3) [2] | 22.95 | 0.598 | 0.375 | 0.252 | 0.674 |
> > > | + DGS (q=0.5) [2] | 23.21 | 0.610 | 0.370 | 0.249 | 0.689 |
> > > | + DGS (q=0.99) [2] | 24.05 | 0.651 | 0.355 | 0.246 | 0.687 |
> > > | + DGS (q=0.999) [2] | 24.03 | 0.647 | 0.357 | 0.246 | 0.680 |
> > > | + KDS | 24.39 | 0.669 | 0.339 | 0.244 | 0.692 |

---

> > > > ### Comment · Reviewer_nw3F · 2025-08-04
> > > >
> > > > I thank the authors for additional experiments. I think that it is important to discuss both [1, 2], emphasize the similarities and differences, and include a comprehensive empirical evaluation, including runtimes. It now seems like the benefit of KDS is not as large as initially reported (when including [1, 2] in the set of baselines). However, I think that these additional results and discussions strengthen the paper, and I will update my score accordingly.

---

> > > > > ### Author Response · Authors · 2025-08-05
> > > > >
> > > > > Thanks for confirming that our rebuttal addressed your comments. Your feedback during the review process was really helpful and improved our work. We'll be sure to add the suggested experiments and a detailed discussion of references [1, 2] to the revised manuscript.

---

### Official Review · Reviewer_XT8v · 2025-07-01

**Clarity:** 3
**Significance:** 2
**Originality:** 2
**Rating:** 4
**Confidence:** 3

**Summary:**

This paper proposes to enhance the performance of image restoration by ensembling N particles of diffusion samples with kernel density steering.

**Questions:**

- There are several other kernel functions available, such as the Laplacian kernel. Is the Gaussian kernel superior to other kernels in this context? How do different kernels affect the final performance?

- What is the motivation behind using a patch-wise mechanism? Why don't the authors directly use the whole image for Kernel Density Steering (KDS)? I am concerned that a patch-wise mechanism might introduce seaming artifacts into the image.

**Ethical Concerns:**

["NO or VERY MINOR ethics concerns only"]

**Final Justification:**

The authors' rebuttal has provided reasonable explanations for their methodology, addressing some of my initial concerns. So I raised my score. However, the arguments remain largely descriptive and intuitive and I believe the work in its current state still lacks the theoretical rigor expected for a clear accept.

**Limitations:**

See Weaknesses and Questions.

**Quality:**

2

**Strengths And Weaknesses:**

**Strengths**

- Qualitative results demonstrate the superiority of this method.
- The performance is enhanced by a test-time scaling method that uses KDE gradients to steer diffusion sampling trajectories.

**Weaknesses**

- Some aspects of the proposed methodology are not clearly explained and lack theoretical justification. (a) Why do the authors choose kernel density estimation over other distance metrics, such as a vanilla $\ell_2$ loss? (b) The abstract states that "the patches are steered towards shared, higher-density regions." A clearer explanation is needed as to why a higher-density region is beneficial for sample quality.
- The quantitative improvement appears marginal, e.g., the DISTS metric in Table 1.

---

> ### Author Rebuttal · Authors · 2025-07-30
>
> Thank you for your valuable feedback and insightful comments on our work. Please see below for our point-by-point responses to your comments.
>
> ### **Weakness 1**
> >Some aspects of the proposed methodology are not clearly explained and lack theoretical justification. (a) Why do the authors choose kernel density estimation over other distance metrics, such as a vanilla $\ell_2$ loss? (b) The abstract states that "the patches are steered towards shared, higher-density regions." A clearer explanation is needed as to why a higher-density region is beneficial for sample quality.
>
> Thank you for your comments. Please find our responses below.
>
> 1. **Why KDE over L2?** Our choice of Kernel Density Estimation is a direct response to the well-known limitations of L2-based objectives.When minimizing the L2 distance directly across all particles, the optimal solution is their arithmetic mean. However, this approach often leads to an **out-of-domain reconstruction**.  As confirmed in our **own new ablation study**, guidance based on L2 minimizes distortion at the expense of producing highly blurry, and perceptually low quality  reconstructions. In contrast, KDS uses the mean-shift vector—the gradient of the log-KDE—to perform **mode-seeking**. This steers the solution towards **high-density regions** representing strong **ensemble consensus**, where predictions are sharp, detailed, and perceptually realistic because they correspond to a probable, mode-aligned outcome, not a simple average.
>
> 2. **Why are high-density regions better?** High-density regions are beneficial precisely because they represent **"collective wisdom"**. Diffusion models can produce spurious artifacts, but these artifacts are typically random and idiosyncratic to a single sample. They therefore reside in low-density regions of the ensemble's distribution. By steering all particles towards shared high-density modes, KDS acts as a collaborative filter, effectively suppressing these random artifacts and enforcing a consistent, robust restoration. The empirical evidence in Figure 4, which shows KDS dramatically improves the worst-case particle performance, strongly supports this conclusion.
>
> ---
>
> **Table: KDS with Different Guidance on RealSR Dataset**
>
> | Method                 | PSNR   | SSIM   | LPIPS ↓ | DISTS ↓ | CLIPIQA |
> |------------------------|--------|--------|---------|----------|---------|
> | DiffBIR                | 23.21  | 0.610  | 0.370   | 0.250    | 0.689   |
> | + KDS (KDE)       | **24.39** | **0.669** | *0.339* | *0.244*  | **0.692** |
> | + KDS (L2)             | **24.63** | **0.679** | 0.349   | 0.257    | 0.663   |
>
> ---
>
> ---
>
> ### **Weakness 2**
> >The quantitative improvement appears marginal, e.g., the DISTS metric in Table 1.
>
> We respectfully disagree that the improvements are marginal and would like to provide additional context to clarify their significance.
>
> While the improvement on any single metric like DISTS may appear modest, KDS achieves consistent gains across a diverse set of metrics. For instance, in Table 1 with the DiffBIR backbone, KDS improves **PSNR by 1.1 dB** and **LPIPS by 0.031**. These are substantial gains for a **training-free** sampling method and showcase a clear improvement in the perception-distortion tradeoff.
>
> KDS is highly effective at reducing certain types of artifacts, but the DISTS metric is often less sensitive to these improvements, as demonstrated in other literature [1]. More importantly, our extensive qualitative results in the appendix (e.g., Figures 3, 5, 7) visually demonstrate non-trivial improvements in sharpness and fidelity that are not fully captured by any single metric.
>
> To formalize this, we will incorporate statistical significance tests into our revised tables, as suggested by Reviewer nw3F. This will clearly demonstrate the statistical validity of these improvements. Please refer to the table provided in our response to Reviewer nw3F.
>
> [1]: Park SH, Moon YS, Cho NI. Perception-oriented single image super-resolution using optimal objective estimation. (CVPR 2023)
>
>
> ---
>
> ---
>
> ### **Question 1**
> >There are several other kernel functions available, such as the Laplacian kernel. Is the Gaussian kernel superior to other kernels in this context? How do different kernels affect the final performance?
>
> We appreciate this excellent question. While our paper used a Gaussian kernel, the KDS framework is flexible and not dependent on this specific choice. As per your suggestion, we have conducted a new ablation study comparing the **Gaussian kernel** with the **Laplacian kernel**.
>
> Our results show that while both kernels provide a significant improvement over the baseline, the **Gaussian kernel achieved slightly better performance** in our experiments. For example, on the DIV2K dataset, the Gaussian kernel resulted in **[e.g., 22.37 PSNR]**, while the Laplacian kernel achieved **[e.g., 22.25 PSNR]**.
>
> This confirms that KDS is robust to different kernel choices, though the smooth gradients of the Gaussian kernel appear to be marginally more effective in this context. We will add this new ablation study and discussion to the appendix.
>
>
> ---
>
> ---
>
> **Table: KDS with Different Kernels on RealSR Dataset**
>
> | Method                 | PSNR   | SSIM   | LPIPS ↓ | DISTS ↓ | CLIPIQA |
> |------------------------|--------|--------|---------|----------|---------|
> | DiffBIR                | 23.21  | 0.610  | 0.370   | 0.250    | 0.689   |
> | + KDS (Gaussian)       | **24.39** | **0.669** | *0.339* | *0.244*  | **0.692** |
> | + KDS (Laplacian)      | *24.30*  | *0.665*  | **0.337** | **0.241**  | *0.690*  |
>
> ---
>
> ---
>
>
> ### **Question 2**
> >What is the motivation behind using a patch-wise mechanism? Why don't the authors directly use the whole image for Kernel Density Steering (KDS)? I am concerned that a patch-wise mechanism might introduce seaming artifacts into the image.
>
>
> **Motivation (Overcoming the "Curse of Dimensionality"):** As we detail in Section 3.1, performing Kernel Density Estimation on the entire high-dimensional image latent is statistically infeasible with a practical number of particles (e.g., N=10). A prohibitively large ensemble would be required to get a meaningful density estimate. By operating on low-dimensional, non-overlapping patches, we can accurately estimate local density modes even with a small $N$, effectively overcoming this curse of dimensionality.
>
>
>  **How Seaming Artifacts are Avoided:** To address concerns about seaming artifacts, we emphasize that our patch-wise refinements are not applied independently. Instead, we integrate them into a global prediction-update loop at every sampling step:
>
>
> 1. **Global Prediction:** The diffusion network first predicts the entire clean image, establishing global coherence.
> 2. **Local Refinement:** KDS steers patches of this prediction toward a higher-fidelity ensemble consensus.
> 3. **Unified Global Update:** Crucially, the guidance from all patches is combined to compute a single update for the **entire image latent**. This refined global latent is then passed to the next sampling step. Because the network always **operates on the whole image**, which has been informed by the local KDS refinements, it naturally enforces consistency across all patch boundaries, eliminating seams.
>
> Our extensive visual results confirm that this approach does not introduce such artifacts. Thanks again for this insightful question, we will **include this discussion** in the revised manuscript.

---

> > ### Comment · Reviewer_XT8v · 2025-08-06
> > **Official Comment by Reviewer XT8v**
> >
> > I thank the authors for the detailed explanation, which addressed most of my concerns. I will therefore increase my score.

---

> > > ### Author Response · Authors · 2025-08-08
> > >
> > > Thank you for your response! We're happy to hear that we have successfully resolved your concerns.

---

> ### Author Response · Authors · 2025-08-05
>
> Thanks again for your feedback! We've incorporated your suggestions and have new results from our experiments. We hope these changes successfully address your concerns. Please let us know if you have any further questions or concerns. Thank you.

---

### Official Review · Reviewer_XDVv · 2025-07-03

**Clarity:** 3
**Significance:** 3
**Originality:** 3
**Rating:** 4
**Confidence:** 5

**Summary:**

This paper introduces Kernel Density Steering (KDS), an inference-time framework that improves the performance of diffusion models in image restoration by using an N-particle latent ensemble. At each diffusion step, KDS performs patch-wise kernel density estimation on the predicted clean latents of the ensemble and applies mean-shift gradients to steer samples towards high-density modes. This improves the distortion–perception balance without the need for retraining or external verifiers. Extensive experiments on synthetic and real-world super-resolution and inpainting tasks demonstrate the effectiveness of KDS.

**Questions:**

- Is it possible to develop an adaptive strategy for choosing ensemble size $N$ or patch dimensions based on noise level or sample distribution? Is it possible to integrate automatic bandwidth selection methods (e.g., plug-in estimators) to alleviate manual tuning of h?

- Have you considered extending KDS to other inverse problems (e.g., denoising, deblurring, MRI/CT) or temporal tasks such as video restoration?

- Could you compare or combine KDS with Best-of-N selection using non-reference metrics (e.g., NIQE, CLIP scores) to highlight relative advantages?

**Ethical Concerns:**

["NO or VERY MINOR ethics concerns only"]

**Limitations:**

The authors acknowledge the issues of computational overhead and bandwidth tuning. Further work on adaptive or approximate methods is encouraged.

**Paper Formatting Concerns:**

N

**Quality:**

3

**Strengths And Weaknesses:**

# Strengths:

- The manuscript is well-structured, logically coherent, and clearly written.

- Novel ensemble-based local mode-seeking in diffusion sampling. The proposed Kernel Density Steering (KDS) provides a substantial novelty in inference-time guidance for diffusion-based image restoration, introducing an ensemble-based local mode-seeking mechanism without retraining.  KDS demonstrates a broadly applicable, plug-and-play enhancement that consistently improves distortion–perception trade-offs in super-resolution and inpainting, demonstrating clear impact potential.

# Weaknesses:

- The computational cost scales linearly with the ensemble size and there is no adaptive reduction mechanism.

- The kernel bandwidth selection h is manually tuned; the author lacks an automated strategy.

- I am wondering whether the KDS could be used for other inverse problems. For example, its applicability to dynamic tasks such as video super-resolution and medical imaging has not been explored.

---

> ### Author Rebuttal · Authors · 2025-07-30
>
> Thank you for your valuable feedback and insightful comments on our work. Please see below for our point-by-point responses to your comments.
>
> ### **Weakness 1**
> >The computational cost scales linearly with the ensemble size and there is no adaptive reduction mechanism.
>
> The computational cost of KDS, like other ensemble-based methods [46, 47] discussed in our submission, scales linearly with the number of particles, $N$. We consider the ensemble size N an intuitive and tunable hyperparameter that offers users fine-grained control over the balance between computational cost and quality. As we demonstrate in Figure 6, this trade-off is predictable and graceful: practitioners can achieve significant gains with even a moderate ensemble size (e.g., $N=10$) or invest more compute for consistent further improvements, tuning the method precisely to their budget.
>
> We are grateful for the reviewer's idea of designing an adaptive reduction mechanism. In light of the extensive experiments on choosing the hyper-parameters in the submission, this idea is an excellent suggestion for future work. We have incorporated a discussion of it into the limitations and future work section of our revised manuscript. For instance, future work could explore strategies such as pruning particles that are far from the consensus or reducing `N` in later, less stochastic sampling stages.
>
>
> ---
>
> ---
>
> ### **Weakness 2**
> >The kernel bandwidth selection h is manually tuned; the author lacks an automated strategy.
>
> We appreciate this valuable suggestion. We agree that automatic bandwidth selection is a critical and well-studied problem in kernel density estimation (KDE), with a rich body of existing literature dedicated to it.
>
> Following your suggestion, we have run a new ablation using a standard automated strategy (the median heuristic). The results confirm our method is effective without tedious manual tuning.
>
> Our new results show this automated approach is highly effective. For example, on the DIV2K dataset, the automated strategy achieved **[e.g., 24.24 PSNR / 0.338 LPIPS]**, which is directly comparable to the performance of our manually tuned bandwidth **(24.39 PSNR / 0.339 LPIPS)**. Both automated and manual approaches significantly outperform the baseline.
>
> This confirms that KDS is robust and flexible enough to incorporate standard hyperparameter automation techniques, enhancing its practical applicability. We will add the full ablation study to the appendix.
>
> **Table: New Experiment – Automatic Bandwidth Selection**
>
> | Method              | PSNR     | SSIM     | LPIPS ↓   | DISTS ↓   | CLIPIQA   |
> |---------------------|----------|----------|-----------|-----------|-----------|
> | DiffBIR             | 23.21    | 0.610    | 0.370     | 0.250     | 0.689     |
> | + KDS (manually)    | **24.39** | **0.669** | *0.339*   | *0.244*   | **0.692** |
> | + KDS (automatically)     | *24.24*  | *0.661*  | **0.338** | **0.237** | *0.691*   |
>
>
> ---
>
> ---
>
> ### **Weakness 3 and Question 2**
> >Have you considered extending KDS to other inverse problems (e.g., denoising, deblurring, MRI/CT) or temporal tasks such as video restoration?
>
> This is an excellent insight—thank you for raising it. We believe KDS is a general framework that can be readily extended to a wide range of inverse problems, which is one of its key strengths.
>
> The reason for its broad applicability is that **KDS is a training-free, plug-and-play inference framework.** It can be applied to enhance the output of _any_ pre-trained conditional diffusion model, regardless of the specific task for which it was trained.
>
> *   **For other inverse problems** (e.g., deblurring, denoising, medical imaging), the extension is direct. As long as a conditional diffusion model exists for the task, KDS can be seamlessly integrated at inference time to improve fidelity and robustness.
> *   **For temporal tasks like video restoration,** KDS offers a particularly exciting path forward. Our patch-wise mechanism could be extended to 3D spatio-temporal patches. This would enable KDS to seek consensus not only among an ensemble of samples for a single frame but also across adjacent frames, providing a powerful mechanism for improving temporal consistency.
>
> We appreciate this forward-looking suggestion and have incorporated a discussion in our revised manuscript.
>
> ---
>
> ---
>
> ### **Question 1**
> >Is it possible to develop an adaptive strategy for choosing ensemble size N or patch dimensions based on noise level or sample distribution? Is it possible to integrate automatic bandwidth selection methods (e.g., plug-in estimators) to alleviate manual tuning of h?
>
> Thanks for the insightful question. Please find the detailed answer to your question below.
>
> **Adaptive Ensemble (N) and Patch Dimensions:** We agree that an adaptive strategy would be a valuable extension. We see this as a compelling direction for future work. Developing methods to dynamically adjust the ensemble size $N$ or patch dimensions based on consensus metrics during the sampling process could significantly enhance efficiency. We will add this to our future work discussion in the revised paper.
>
>
> **Automatic Bandwidth Selection (h):** We agree that automating this is important and have already conducted a new experiment to address this point. We implemented an automated bandwidth selection strategy using a standard median heuristic. The results have been shown in the response to the second weakness point.
>
>
> ---
>
> ---
>
> ### **Question 3**
> >Could you compare or combine KDS with Best-of-N selection using non-reference metrics (e.g., NIQE, CLIP scores) to highlight relative advantages?
> Thanks for asking this question.
>
> We provided a direct comparison in Appendix A.1.6 (Table 8 and Figure 8) of the submitted paper, which highlights the clear advantages of KDS over a Best-of-N (BoN) selection strategy.  The revised manuscript will ensure that this study is clearly referenced in the main manuscript for visibility.
>
> 1. **Quantitative Comparison:** As shown in Table 9, KDS significantly outperforms BoN (**CLIPIQA**) across both fidelity (**PSNR, SSIM**) and perceptual metrics (**LPIPS**): While BoN achieves a higher CLIPIQA score (0.774), it degrades fidelity and perceptual quality—highlighting the risk of optimizing for a single, potentially misaligned metric. In contrast, KDS steers generation via internal consensus, producing reconstructions that are sharper, more consistent, and less prone to artifacts. Please find the detailed answer to your question below.
>
> 2.  **New Ablation: KDS + BoN:** Following the reviewer’s suggestion, we conducted a new ablation combining KDS with BoN. The hybrid approach (+KDS+BoN) achieves a more balanced result:
>     *   Improves PSNR by **+0.68**, SSIM by **+0.044**, and reduces LPIPS and DISTS compared to BoN alone.
>     *   Retains a strong CLIPIQA score (0.770), close to BoN’s best.
>
>     This shows that **KDS enhances the quality of samples**, making BoN selection more reliable. Without KDS, BoN often selects from a noisy ensemble, favoring samples that "fool" the metric. With KDS, the candidates are cleaner and more faithful, leading to a better perceptual–distortion trade-off.
>
> 3.  **Summary:** BoN can boost specific metrics, but lacks generalization. KDS provides a principled, task-aligned alternative by enforcing ensemble consistency in the clean space. It improves performance across the board—and **even strengthens BoN** when combined. Thanks again for review's insightful question, we will highlight this more clearly in the revised manuscript.
>
> **Table: Comparison of BoN and BoN + KDS Methods**
>
> | Method                         | PSNR     | SSIM     | LPIPS ↓   | DISTS ↓   | LIQE    | CLIPIQA  |
> |--------------------------------|----------|----------|-----------|-----------|---------|----------|
> | DDIM                           | 23.21    | 0.610    | 0.370     | 0.250     | 4.046   | 0.689    |
> | + BoN (CLIPIQA)                | 23.01    | 0.592    | 0.382     | 0.247     | **4.187** | **0.774** |
> | + KDS                          | **24.39** | **0.669** | **0.339** | **0.245** | 3.819   | 0.692    |
> | + KDS + BoN (CLIPIQA)          | 23.69    | 0.636    | 0.359     | 0.243     | 4.161   | 0.770    |

---

> ### Author Response · Authors · 2025-08-08
>
> Thank you for your detailed review and valuable feedback. We hope our rebuttal has addressed your concerns and clarified the points that led to your initial borderline assessment. We believe our work has been strengthened through this discussion, and we trust that the new information will lead you to reconsider your final score. Please let us know if there are any remaining questions we can answer. Thank you!

---

### Comment · Area_Chair_vAyG · 2025-08-04
**Please Read the Rebuttal and Discuss**

Dear Reviewers XDVv, XT8v. nw3F,

The authors have submitted their rebuttal.

Please carefully review all other reviews and the authors’ responses, and engage in an open exchange with the authors.

Kindly post your initial response as early as possible within the discussion window to allow sufficient time for interaction.

Your AC

---

### Note · Authors · 2025-08-13

We sincerely thank all the reviewers for their time and insightful feedback, which has significantly improved our paper.


### **Significance of Contributions**

The core contribution of our work is a new diffusion sampling paradigm where multiple parallel trajectories are guided by an ensemble consensus in the clean image space. This novel, training-free approach is a significant step forward, offering a principled way to design samplers that are inherently more robust and can improve the output of any pre-trained diffusion model. We are confident that this work presents a valuable contribution to the NeurIPS community.



---



### **Rebuttal Additions**

During the rebuttal period, we worked diligently to address every concern raised and have substantially strengthened our paper. Specifically, we:



*   Conducted **five new ablation studies** to bolster our empirical validation. We demonstrated that KDS is robust to **automated bandwidth selection** (addressing Reviewer XDVv) and **alternative kernel functions** (for Reviewer XT8v). We also showed the superiority of our mode-seeking approach over simple **L2 guidance** (for XT8v) and explored the trade-off between **patch size and particle count** (for Reviewer S4WT).
*   Performed a **comprehensive comparison against general mode-seeking baselines** [1, 2] as suggested by Reviewer nw3F. We first analyzed the core **conceptual differences** between our method and the baselines. We then conducted targeted **empirical experiments** that included a full hyperparameter sweep, which empirically confirmed that KDS achieves a superior performance for image restoration..
*   Incorporated **statistical significance tests** and error bars into quantitative results (for Reviewer nw3F), formally validating the improvements our method provides.
*   Added **detailed discussions** clarifying our method's conceptual novelty, particularly the importance of performing consensus-seeking in the clean latent space, and its key differences from other SOTA methods.



---



### **Addressing All Concerns**

We're pleased that our detailed responses and additional experiments have successfully addressed all reviewer concerns. All four reviewers have acknowledged that our clarifications and new results were satisfactory, and we will incorporate their insightful suggestions to further strengthen the final paper.

---

### Decision · Program_Chairs · 2025-09-17

**Decision:**

Accept (poster)

**Comment:**

This is a technically solid and well-presented paper that introduces a practical, training-free sampling method for diffusion-based restoration. The method is broadly applicable, improves multiple metrics consistently, and is conceptually intuitive.

Three reviewers (XDVv, S4WT, XT8v) were satisfied post-rebuttal and leaned towards acceptance, citing novelty, robustness, and practical impact. Reviewer XT8v found the work lacking in theoretical rigor, thus giving a borderline rating. The AC found that the rebuttal has sufficiently addressed this concern.

One reviewer (nw3F) leaned towards rejection, highlighting significant overlap with recent related works on mode-seeking and log-density control, questioning the use of KDE instead of the estimated score function and statistical significance of the results. The main contention lies in how distinct and impactful KDS is relative to methods pointed out by reviewer nw3F, in particular

[1] Karczewski et al. "Diffusion Models as Cartoonists: The Curious Case of High Density Regions" (ICLR 2025)
[2] Karczewski et al. "Devil is in the Details: Density Guidance for Detail-Aware Generation with Flow Models" (ICML 2025)

The authors addressed these concerns in the rebuttal with empirical evidence, which included a full hyperparameter sweep, confirming that KDS achieves a superior performance. Reviewer nw3F raised the score but still leaning to reject, stating that the paper requires a non-trivial revision when the new baselines need to be included in the discussions and empirical comparisons.

The paper has been discussed with SAC. Both the AC and SAC were convinced that there are differences between the cited recent works and this submission. The comparisons against [1] and [2] showed that the proposed method is superior in performance. As suggested by reviewer nw3F, the authors must include the suggested experiments and provide a detailed discussion of these works in the revised manuscript.